# VCP maintains nuclear size by regulating the DNA damage-associated MDC1–p53–autophagy axis in *Drosophila*

Ya-Chu Chang [1,2], Yu-Xiang Peng[1,5], Bo-Hua Yu[1,5], Henry C. Chang[3], Pei-Shin Liang[1], Ting-Yi Huang[1], Chao-Jie Shih[1], Li-An Chu [2,4] & Tzu-Kang Sang[1,2 ✉]

The maintenance of constant karyoplasmic ratios suggests that nuclear size has physiological significance. Nuclear size anomalies have been linked to malignant transformation, although the mechanism remains unclear. By expressing dominant-negative TER94 mutants in *Drosophila* photoreceptors, here we show disruption of VCP (valosin-containing protein, human TER94 ortholog), a ubiquitin-dependent segregase, causes progressive nuclear size increase. Loss of VCP function leads to accumulations of MDC1 (mediator of DNA damage checkpoint protein 1), connecting DNA damage or associated responses to enlarged nuclei. TER94 can interact with MDC1 and decreases MDC1 levels, suggesting that MDC1 is a VCP substrate. Our evidence indicates that MDC1 accumulation stabilizes p53A, leading to TER94[K2A]-associated nuclear size increase. Together with a previous report that p53A disrupts autophagic flux, we propose that the stabilization of p53A in TER94[K2A]-expressing cells likely hinders the removal of nuclear content, resulting in aberrant nuclear size increase.

[1] Institute of Biotechnology, Department of Life Science, National Tsing Hua University, Hsinchu, Taiwan. [2] Brain Research Center, National Tsing Hua University, Hsinchu, Taiwan. [3] Department of Biological Sciences, Purdue University, West Lafayette, IN, USA. [4] Department of Biomedical Engineering and Environmental Science, National Tsing Hua University, Hsinchu, Taiwan. [5] These authors contributed equally: Yu-Xiang Peng, Bo-Hua Yu. ✉email: tksang@life. nthu.edu.tw

It has long been appreciated that cells keep a relatively constant karyoplasmic ratio, i.e., the proportion of nuclear volume and cytoplasmic volume[1], implying maintaining such a physiological consistency has important roles. Indeed, aberrant nuclear size has served as a pathological hallmark in several types of malignant tumors[2,3]. However, it is unclear whether the change in nuclear size has a direct role in causing diseases or it represents a byproduct of another cellular defect that causes diseases. To address this, it is critical to understand the regulatory processes, as well as the cellular stresses, that impact nuclear size.

Studies from multiple systems have revealed that nuclear size is proportionally scaled to the cytoplasmic volume[1,4–9]. This reliance on cytoplasmic volume suggests that nucleus can sense determinants for nuclear size control residing outside the nucleus. The import of nuclear lamina proteins through the nuclear pore complexes (NPCs) has been implicated in the nucleus growth during interphase[10,11]. The availability of ER membrane, the source for reassembled nuclear envelope at the end of cell division[12], also constrains the nucleus size and shape after mitosis[13]. Recent genetic screens have further implicated nucleocytoplasmic transport, RNA processing, and nucleo-cytoskeletal interactions as key mechanisms for controlling nucleocytoplasmic ratio[14,15]. Cellular stresses like DNA damages can influence transcription and cause accumulation of certain nuclear proteins, although the impact of DNA lesions on nuclear size has not been thoroughly examined. In this study, we investigate the molecular mechanisms linking nuclear size control and DNA damage using *Drosophila* TER94, the fly homolog of VCP, which is known to function in both DNA damage response (DDR) and nuclear envelope reassembly after mitosis (see below).

We have previously used *Drosophila* to model IBMPFD (Inclusion body myopathy with early-onset Paget disease and frontotemporal dementia), a multi-system degenerative disorder associated with specific mutations in VCP[16]. VCP ATPases form hexamers and extract ubiquitinated proteins from organelles and protein complexes for proteasome-mediated degradation, re-folding, or liberation[17]. VCP has been implicated in numerous processes. In the cytoplasm, VCP responds to stresses from the endoplasmic reticulum (ER) by participating in the ER-associated degradation (ERAD) of misfolded ER proteins to maintain the homeostasis[18]. A similar requirement for VCP function has also been reported for removing damaged proteins during mitochondrial quality control[19]. VCP is essential for the maturation of autophagosomes and their subsequent fusion with lysosomes[20,21]. During mitotic division, the nuclear envelope and Golgi reassembly also require VCP[22,23]. In the nucleus, this ubiquitin-dependent protein segregase/unfoldase has been reported to remove or degrade several chromatin-associated proteins involved in DDR[24]. With our *Drosophila* model, we show that IBMPFD alleles affect energy consumption[16]. Whether *Drosophila* TER94 has a nuclear function is not clear.

Machinery repairing DNA lesions operates with a precise spatiotemporal sequence, completing the tasks of lesion sensing, signaling, and repairing. To achieve this, DDR proteins recruited to the DNA damage sites for one task need to be displaced to allow the assembly of complexes for a different task. This dynamic regulation involves several post-translational modifications, including phosphorylation, ubiquitination, and SUMO (small ubiquitin-like modifier)-conjugation[25]. For instance, in response to double-strand breaks, the MRN (MRE11-RAD50-NBS1) sensor complex is loaded onto the damage sites, facilitating the recruitment of phosphatidylinositol-3 kinase family protein ATM (known as Tefu in fly) and ATM-dependent phosphorylation of the histone variant H2AX (known as H2AV in fly)[26]. The phosphorylated H2AX (γH2AX, known as γH2AV in *Drosophila*) then recruits MDC1 (known as Mu2 in fly), which

serves as an adaptor for additional factors to remodel chromatins near the DNA damage sites[27]. A SUMO-targeted E3 ligase RNF4 (known as Dgrn in fly) mediates the ubiquitination of MDC1, which promotes its subsequent dissociation from chromatin[28]. The removal of MDC1 is required for the recruitment of downstream signaling and repair proteins, such as 53BP1 and BRCA1[28,29]. However, how ubiquitin-conjugated MDC1 are removed from chromatin is not understood.

In this study, we present evidence that loss of TER94 function leads to nuclear size increase. Consistent with the many known VCP nuclear substrates[30], ubiquitinated proteins accumulate inside the nucleus of TER94 dysfunction cells, and the extent of accumulation correlates positively with the degree of nuclear enlargement. This TER94-associated nuclear size increase is modified by Mu2 RNAi and overexpression, linking DDR to this process. Genetic evidence further suggests the p53A isoform acts downstream of TER94 and Mu2 in this aberrant nuclear size increase. TER94 interacts physically with Mu2 and negatively regulates Mu2 level, suggesting Mu2 is a substrate of TER94. Mu2 positively regulates p53 protein level and is capable of directly interacting with p53. In addition, TER94 dysfunction, in a Mu2- and p53-dependent manner, causes accumulation of p62, a well-known indicator of disrupted autophagic flux. Together, our findings suggest that TER94 dysfunction disrupts DDR, causing an accumulation of Mu2 proteins, which stabilize p53 and perturb autophagy to enlarge the nuclei.

## Results

**Disruption of TER94 function causes nuclear enlargement in postmitotic cells.** VCP is an essential gene, and whole-animal knockouts of VCP function in both fly and mouse cause embryonic lethality[31,32]. Thus, to analyze the cellular roles of VCP in adults, we chose an alternative approach of expressing UAS-controlled dominant-negative transgenes with tissue-specific GAL4 drivers. We have previously generated UAS-TER94^K2A (K248A & K521A) and UAS-TER94^E2Q (E302Q & E575Q), which overexpress *Drosophila* VCP mutants affecting ATP-binding and ATP-hydrolysis, respectively[33,34]. These variants with disabled ATPase activity are thought to act as dominant-negatives by forming non-functional hexamers with endogenous wild-type TER94, thereby depleting the functional TER94 pool[18,35]. When expressing TER94^K2A with *Rh1-GAL4* (hereafter referred as *Rh1 > TER94^K2A*), a driver active in postmitotic outer photoreceptor cells R1-R6 (R cells, Fig. 1a) from the late pupal stage on, we observed an age-dependent increase in the nuclear size (Supplementary Fig. 1). We measured the R cell nuclear sizes by quantifying antibody-labeled nuclear areas in confocal sections and normalizing them to the cross-section areas of rhabdomeres, cylindrical photo-sensory organelles present in all R cells, from the same cells. As revealed by anti-Lamin antibody staining, which labels the nuclear lamina, the R cell nucleus transforms from a fringe-like shape (Supplementary Fig. 1a, 70% pupal) to a round contour (Supplementary Fig. 1a, 0-day adult) at the pupae-to-adult transition. This nuclear morphology transformation was similar in *Rh1 > LacZ* control and *Rh1 > TER94^K2A*, indicating that expression of dominant-negative TER94 does not hinder nuclear maturation during development. Instead, *Rh1 > TER94^K2A* nuclei in 2-day-old adults, compared to those of *Rh1 > LacZ*, exhibited a 35% increase in size (Supplementary Fig. 1). From this point on, while *Rh1 > LacZ* nuclear size remained constant, the *Rh1 > TER94^K2A* nuclei continued to enlarge, as they were 117% larger than the control in 5-day-old adults (Fig. 1b, c and Supplementary Fig. 1), and 129% larger than the control in 8-day-old adults (Supplementary Fig. 1).

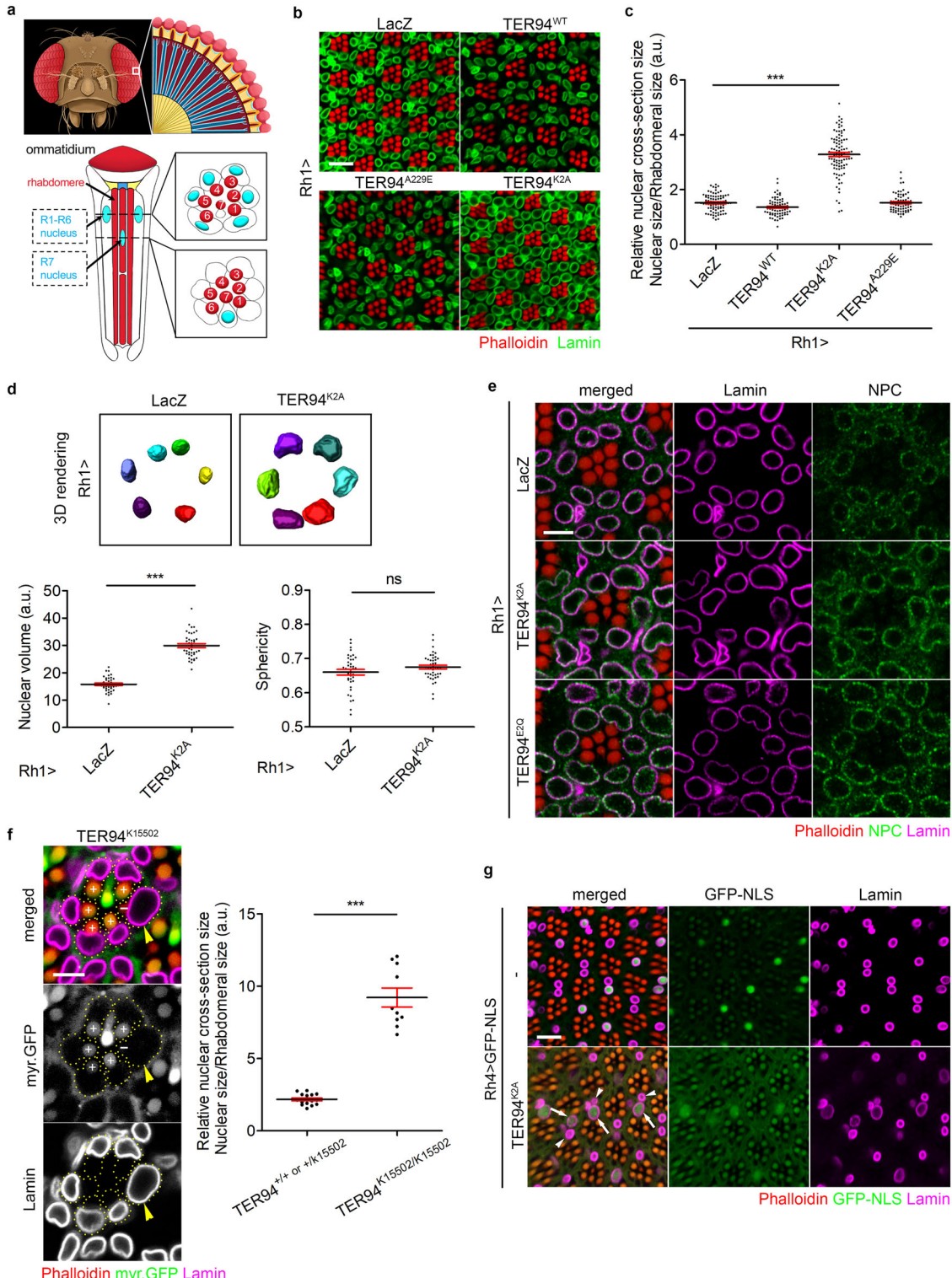

To ensure these measurements from single confocal images accurately reflect the nuclear sizes, we compared the volumes of 3D representations of photoreceptor nuclei, reconstructed from Z-stacks of anti-Lamin sections (Supplementary Movie 1). The analysis revealed that in 5-day-old adults, TER94$^{K2A}$ expression, while having no effect on the nucleus shape (Fig. 1d), approximately doubled the nuclear volume (Fig. 1d). *Rh1 > TER94$^{K2A}$* stained with anti-NPC (nuclear pore complex) antibody yielded nuclear enlargement as well, indicating the above-mentioned phenotype was not caused by Lamin mis-localization

(Fig. 1e). This expanded nuclear size phenotype is specific, as the cell size revealed with a plasma membrane-associated mCherry and rhabdomere size remained unaltered by TER94$^{K2A}$ expression (Supplementary Fig. 2). Together, these results showed that TER94$^{K2A}$ expression causes a post-development and progressive nuclear size enlargement.

This expansion of nuclear size was not seen in cell overexpressing *TER94$^{A229E}$* (Fig. 1b, c), an IBMPFD-associated variant, which we have previously shown to act as a dominant-active mutant[16], suggesting that aberrant nuclear enlargement is

**Fig. 1 Loss of TER94 function induces nuclear expansion. a** A cartoon depiction of *Drosophila* compound eye. The photon-detecting rhabdomere and the R1-R7 nuclei are indicated. Two dash lines mark cross-section of R1-R6 and R7 nuclei with the corresponding rhabdomeres. **b** Confocal images of 5-day-old adult retinas expressing indicated transgenes under the control of *Rh1-GAL4* stained with phalloidin (red) and anti-Lamin (green) antibody. **c** Quantification of cross-section area of R1-R6 nuclei from flies of indicated genotypes. The number of independent nuclei measured are 84 (LacZ), 72 (TER94$^{WT}$), 96 (TER94$^{K2A}$), and 67 (TER94$^{A229E}$). Values shown are arbitrary units (a.u.) representing mean (black line) ± SE (red line). One-way ANOVA with Bonferroni's multiple comparison test, ***$p < 1.0e-15$ compared to $Rh1 > LacZ$. **d** Top: 3D-rendering of R1-R6 nuclei from $Rh1 > LacZ$ and $Rh1 > TER94^{K2A}$ unit eyes. Individual nuclei are color-coded. Bottom: Quantitative comparison of R1-R6 nuclear volume and sphericity from $Rh1 > LacZ$ ($n = 39$) and $Rh1 > TER94^{K2A}$ ($n = 40$/volume and 43/sphericity). Values shown represent mean ± SE (Student's *t* test, two-tailed, nuclear volume: ***$p < 1.0e-15$; sphericity: $p = 0.1369$, ns not significant). **e** Confocal images of 5-day-old adult retinas expressing indicated transgenes stained with phalloidin (red), anti-Lamin (magenta), and anti-NPC (green). **f** Left: An adult eye bearing TER94$^{K15502}$ clone is stained with phalloidin (red) and anti-Lamin (magenta) antibody. One homozygous TER94$^{K15502}$ cell (arrowhead), marked by the absence of myr.GFP in the rhabdomere (-), contains enlarged nucleus. Dash lines outline R cell boundaries. Right: Quantification of the nuclear cross-section area of normal (+/+) or TER94 heterozygous (+/TER94$^{K15502}$) versus TER94$^{K15502}$ cells is shown. The number of independent nuclei measured are 15 (+/+ or +/TER94$^{K15502}$) and 10 (TER94$^{K15502}$). Student's *t* test, two-tailed, ***$p = 5.3e-12$. **g** Confocal images of 5-day-old adult retinas expressing GFP-NLS (green) alone (-) or co-expressing GFP-NLS and TER94$^{K2A}$ stained with phalloidin (red) and anti-Lamin (magenta). The R7 photoreceptors with active *Rh4-GAL4* are GFP-NLS labeled. The arrows indicate the enlarged nuclei of R7 expressing TER94$^{K2A}$, and the arrowheads indicate the normal nuclei. The number of independent experiments performed: 2 (**e**), 6 (**g**). Scale bars: 10 μm (**b, g**), 5 μm (**e, f**).

---

not a cause to this dementia. Overexpression of $TER94^{E2Q}$, a different dominant-negative allele, also caused nuclear enlargement, indicating that this phenotype was not specific to the $TER94^{K2A}$ variant (Fig. 1e). To further confirm that a loss of TER94 function causes this nuclear size aberrance, we generated mosaic eyes containing mutant clones of $TER94^{K15502}$ (a strong hypomorphic allele), and showed that homozygous $TER94^{K15502}$ cells, marked by the absence of GFP, exhibited nuclei larger than those of neighboring normal cells (Fig. 1f).

To ensure the nuclear size increase is not restricted to the outer R cells, we used *Rh4-GAL4* driver, which is active in ~70% of R7 cells (Fig. 1a), to express *UAS-TER94$^{K2A}$* and *UAS-GFP-NLS* (*UAS-GFP-NLS* was included to identify the $TER94^{K2A}$-expressing R7 population). Consistent with the $Rh1 > TER94^{K2A}$ results, R7 nuclei expressing TER94$^{K2A}$ (Fig. 1g, arrows) were larger than those without TER94$^{K2A}$ expression (Fig. 1g, arrowheads). Expressing TER94$^{K2A}$ in other non-neuronal postmitotic cells, such as cone cells and indirect flight muscle cells, recapitulated this aberrant increase of nuclear size (Supplementary Fig. 3). Altogether, these data suggest that TER94 AAA ATPase has a fundamental role in regulating nuclear size in postmitotic cells.

**Loss of TER94 function causes accumulations of autophagic markers and ubiquitinated nuclear proteins.** VCP hexamer is known to process ubiquitinated substrates for degradation in the ubiquitin-proteasome system (UPS), and has been linked to autophagy, DDR, ERAD, and apoptosis[17]. To determine whether disruption of any of the aforementioned processes contributes to the nucleus expansion, we monitored the reporter activities for these pathways in $Rh1 > TER94^{K2A}$ retina. Given that TER94 targets ubiquitinated substrates and ubiquitin-conjugation is required by both UPS and autophagy, we stained $Rh1 > TER94^{K2A}$ adult retina with FK2, an antibody that recognizes mono- and polyubiquitinated conjugates[36]. At the late-pupal stage, at which the nuclear size of $Rh1 > TER94^{K2A}$ outer R cells appeared normal, FK2 signal was seen in the nuclei of only a few cells (Fig. 2a). As the flies aged, the FK2 signal intensity and number of cells with nuclear FK2 staining increased, suggesting that $TER94^{K2A}$ expression causes an accumulation of ubiquitinated substrates. In comparison, no nuclear FK2 signal and only occasional cytoplasmic FK2-positive puncta were detected in $Rh1 > LacZ$ (Fig. 2a). This accumulation of ubiquitinated nuclear proteins was caused by loss of TER94 function, confirmed by the FK2 immunostaining of mosaic tissues bearing homozygous $TER94^{K15502}$ clones (Fig. 2c). Importantly, the strength of

FK2 signal correlated strongly with the nuclear size increase (Fig. 2a, b). This correlation, along with the fact that the appearance of FK2 staining precedes nuclear enlargement, suggests the failure in removing ubiquitinated nuclear proteins in $Rh1 > TER94^{K2A}$ cells is a cause for the aberrant nuclear expansion.

In 2-day-old $Rh1 > TER94^{K2A}$ adults, while a significant portion of nuclei were already enlarged, the activity of CD3δ-YFP, an ERAD reporter requires UPS degradation, remained low, implying that ERAD/UPS impairment is unlikely to be a cause (Supplementary Fig. 4a). The signal of CD8-PARP-Venus, an apoptotic reporter, was absent in 2-day-old $Rh1 > TER94^{K2A}$ adults and could only be detected in a few $Rh1 > TER94^{K2A}$ cells 1 week after the onset of nuclear expansion (Supplementary Fig. 4b). Similarly, another independent apoptotic reporter, cleaved Caspase-3, was appeared only in a subset of 5-day-old R7 cells expressing TER94$^{K2A}$ by *Rh4-GAL4* (Supplementary Fig. 4c). These observations preclude the notion of apoptosis being a cause for enlarged nuclei.

To test whether TER94$^{K2A}$ affects autophagy, we monitored the level of GFP-Atg8a and GFP-p62 in $Rh1 > TER94^{K2A}$. Atg8a is essential for autophagosome formation[37] and p62 serves as an adaptor linking ubiquitinated substrates to Atg8[38]. In control retina, GFP-Atg8a and GFP-p62 signals were low and remained unchanged as the flies aged (Fig. 2d, e). In contrast, both GFP-Atg8a and GFP-p62 showed age-dependent increase in $Rh1 > TER94^{K2A}$ (Fig. 2d, e). Immunolabeling with anti-LC3 and anti-ref(2)p, respectively, confirmed the accumulation of endogenous Atg8 and p62 in TER94 dysfunction cells (Supplementary Fig. 5). The GFP-Atg8a increase was both cytoplasmic and nuclear, and correlated well with cells of large nuclei (Fig. 2d). While the GFP-p62 increase also showed some correlation with cells of large nuclei, its signals were more spatially restricted and age-dependent (Fig. 2e). Large GFP-p62 positive aggregates were seen primarily in the cytoplasm in 2-day-old retina, whereas these structures were observed in both the cytoplasm and nuclei in 5-day-old retina (Fig. 2e). Both autophagy GFP-tagged reporters were expressed from a heterologous promoter (UAS), suggesting that the observed increases are mediated by post-transcriptional mechanisms. Furthermore, Atg8 and p62 are degraded after autophagosome–lysosome fusion under normal condition, and a prolonged accumulation of these two proteins has been associated with autophagy deregulation[39]. The fact that $Rh1 > TER94^{K2A}$ showed age-dependent accumulation of Atg8a and p62 suggests defective autophagy could be a contributing factor for the nuclear enlargement.

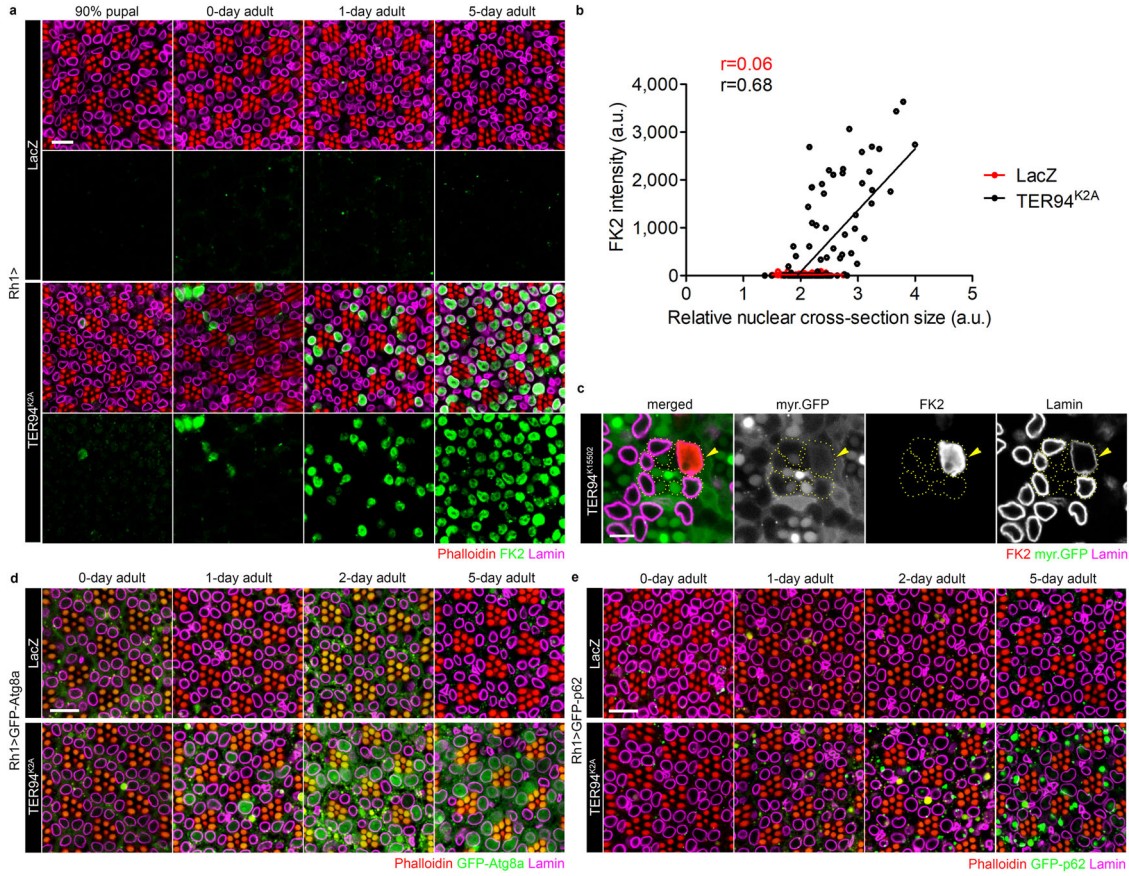

**Fig. 2 Atg8, p62, and ubiquitinated nuclear proteins accumulate in cells lacking TER94 function. a** Time-course analysis of the nuclear size change and the presence of ubiquitinated proteins in *Rh1 > LacZ and Rh1 > TER94^K2A* pupal or adult eyes stained with phalloidin (red), anti-Lamin (magenta), and anti-Ubiquitin conjugates (FK2, green) antibodies. **b** Pearson's correlation analysis of nuclear cross-section size and FK2 intensity from images of 1-day-old adult retina (as shown in panel **a**). The number of independent nuclei measured are 88 (LacZ) and 125 (TER94^K2A). *r* values represent Pearson's correlation coefficient in each group. **c** An adult *TER94^K15502* clone, marked by the absence of myr.GFP (arrowhead), stained with anti-Lamin (magenta) and anti-Ubiquitin conjugates (FK2, red) antibodies. R cell boundaries are outlined in dash lines. **d, e** Retinas from freshly eclosed (0-day) to 5-day-old adults co-expressing GFP-Atg8a (**d**) or GFP-p62 (**e**) with LacZ (control) or TER94^K2A under the control of *Rh1-GAL4* stained with phalloidin (red) and anti-Lamin (magenta) antibody. The number of independent experiments performed: **a** 2 (LacZ), 5 (TER94^K2A) for 90% pupal; 3 (LacZ), 2 (TER94^K2A) for 0-day; 4 (LacZ), 6 (TER94^K2A) for 1-day; 6 (LacZ), 6 (TER94^K2A) for 5-day adult. **c** 6. **d** 3 (0-day), 2 (1-day), 6 (2-day), 3 (5-day). **e** 2 (0-day), 5 (1-day), 5 (2-day), 4 (5-day). Scale bars: 10 μm (**a**, **d**, **e**), 5 μm (**c**).

**Disruption of nuclear TER94 function in DNA damage repair affects nuclear size.** VCP has known functions in both the cytoplasm and nucleoplasm[17]. Given the correlation between ubiquitinated protein accumulation in the nucleus and the nuclear expansion, we speculated that these were due to a deficit of TER94 function in the nucleus. To test this hypothesis, we took advantage of a fortuitous finding of ours that overexpression of Derlin-1, an ER membrane protein that recruits TER94 to the ER vicinity upon ER stress, depletes the nuclear pool of endogenous TER94[40] (Fig. 3a, b). Indeed, overexpression of Derlin-1 caused nuclear expansion (Fig. 3a, b), supporting the notion that this nucleus enlargement was caused by a depletion of nuclear TER94. As a control, expression of Derlin-1^ΔSHP, a Derlin-1 construct incapable of binding to TER94[40], had no effect on the nuclear TER94 level and the nuclear size (Fig. 3a, b). Furthermore, co-expression of wild-type TER94 in *Rh1 > Derlin-1* suppressed the nuclear expansion phenotype (Supplementary Fig. 6), presumably by replenishing the level of TER94 in the nucleus.

Within the nucleus, VCP function has been implicated in cell cycle and DDR regulations[41,42]. Using EdU incorporation assay, *Drosophila*-specific FUCCI (Fluorescent Ubiquitination-based Cell Cycle Indicator) system (Fly-FUCCI)[43], and another cell cycle reporter S/G2/M-Green, we showed that *Rh1 > TER94^K2A*

cells remained mitotically inactive (Supplementary Fig. 7), indicating that the nuclear morphological change is not caused by cell cycle re-entry. To test the impact of TER94 inactivation on DDR, we stained *Rh1 > TER94^K2A* for γH2AV, a phosphorylated histone H2A variant formed at DNA damage sites to facilitate DDR processing[44,45]. Like in mammals, robust γH2AV signals react to genotoxic insults in flies, as tested by $H_2O_2$ treatment (Fig. 3c). In the absence of external mutagen treatment, this DDR marker signal remained unchanged in *Rh1 > LacZ* cells, whereas γH2AV staining increased in the nuclei of TER94^K2A-expressing cells (Fig. 3d, e). More importantly, the elevation of γH2AV staining in *Rh1 > TER94^K2A* correlated with the onset of nuclear size expansion. Like the nuclear enlargement phenotype, increased γH2AV staining was detected in homozygous *TER94^K15502* clones in mosaic eyes (Fig. 3f).

To exclude the possibility that *Rh1 > TER94^K2A* merely altered γH2AV expression, we showed that γH2AV transiently increased in wild-type adult retina with acute exposure to γ-radiation (Supplementary Fig. 8a, b), confirming γH2AV's specificity in detecting DNA lesions, as well as reflecting the cells' ability to quickly repair these induced mutations. In contrast, *Rh1 > TER94^K2A* exposed to γ-radiation showed sustained γH2AV increases (Supplementary Fig. 8c, d), indicative of a disruption in

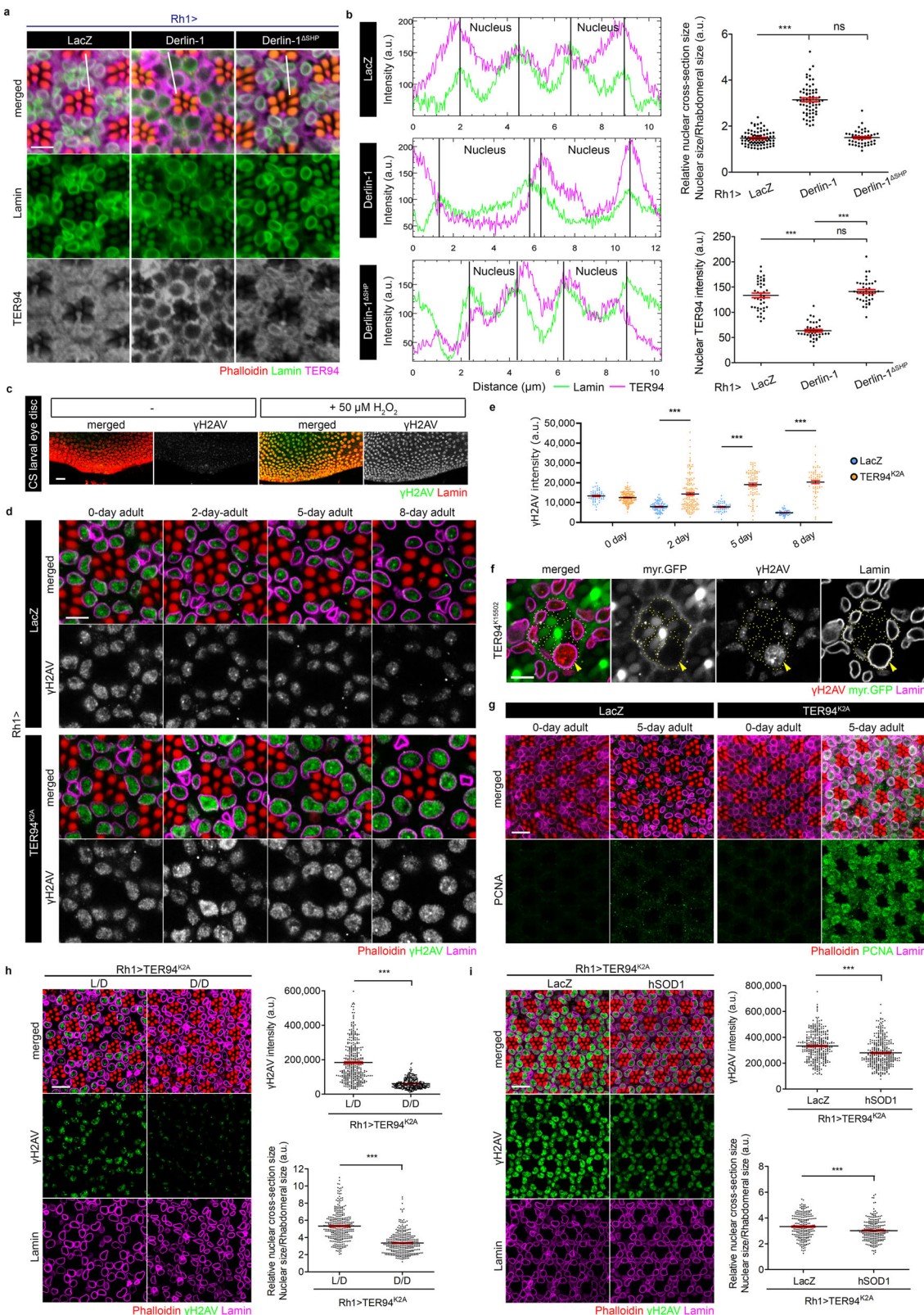

the cells' ability to repair DNA damages. In addition, we tested another DDR marker, PCNA (Proliferating Cell Nuclear Antigen), a DNA polymerase δ processivity factor essential for both DNA replication and repair[46], and observed an elevation of PCNA in *Rh1 > TER94^K2A* (Fig. 3g). As we have excluded the possibility of *Rh1 > TER94^K2A* initiating DNA replication, this PCNA observation, along with the abovementioned γH2AV

results, indicates that disruption of TER94 function perturbs the DDR process, possibly causing an accumulation of unrepaired DNA lesions.

To pinpoint the source of these unrepaired DNA lesions, we monitored the level of γH2AV in *Rh1 > TER94^K2A* flies raised in the dark, which minimizes their exposure to radiation and UV light. Compared to *Rh1 > TER94^K2A* grown in the regular 12 h light/dark

**Fig. 3 Blocking TER94 function hampers DNA damage repair. a** Confocal images of 5-day-old $Rh1 > LacZ$, $Rh1 > Derlin-1$, and $Rh1 > Derlin-1^{\Delta SHP}$ retinas stained with phalloidin (red), anti-Lamin (green), and anti-TER94 (magenta). **b** Left: Analysis of TER94 subcellular distribution. The analyzed range is indicated by the white lines shown in panel **a**. Image J measures the intensities of anti-Lamin and anti-TER94 signals along the lines. The peaks of the anti-Lamin signal define the borders of the nuclei. Right: Quantification of the cross-section area of R1-R6 nuclei (top) and the nuclear TER94 intensity (cumulated pixel of TER94 along the line divided by the line distance defined by anti-Lamin-marked two peaks, bottom). **c** Confocal images of wild-type larval eye discs with or without 50 μM $H_2O_2$ treatment stained with anti-Lamin (red) and anti-γH2AV (green). **d** Time-course analysis of the change in nuclear size and the level of γH2AV from $Rh1 > LacZ$ and $Rh1 > TER94^{K2A}$ adult eyes stained with phalloidin (red), anti-Lamin (magenta), and anti-γH2AV (green). **e** Quantification of the anti-γH2AV signal intensity over time from $Rh1 > LacZ$ and $Rh1 > TER94^{K2A}$. **f** An adult $TER94^{K15502}$ clone, marked by the absence of myr.GFP (arrowheads), stained with anti-Lamin (magenta) and anti-γH2AV (red). R cell boundaries are outlined in dash lines. **g** Confocal images of $Rh1 > LacZ$ and $Rh1 > TER94^{K2A}$ adult retinas stained with phalloidin (red), anti-Lamin (magenta), and anti-PCNA (green). **h** Left: Confocal images of 5-day-old $Rh1 > TER94^{K2A}$ retinas from flies raised in normal 12 h light/12 h dark (L/D) or complete dark (D/D) conditions after birth stained with phalloidin (red), anti-Lamin (magenta), and anti-γH2AV (green). Right: Quantification of the γH2AV intensity (top) and the cross-section area of R1-R6 nuclei (bottom). **i** Left: Confocal images of 5-day-old retinas from $Rh1 > TER94^{K2A}$ co-expressing either $LacZ$ or $hSOD1$ stained with phalloidin (red), anti-Lamin (magenta), and anti-γH2AV (green). Right: Quantification of the γH2AV intensity (top) and the cross-section area of R1-R6 nuclei (bottom). The number of independent experiments performed: 2 (**a**), 4 (**c**), 6 (**f**), 4 (**g**). Quantification details of **b**, **e**, **h**, and **i** are listed in Statistics and reproducibility. Scale bars: 5 μm (**a**, **d**, **f**), 20 μm (**c**), 10 μm (**g**, **h**, **i**).

cycle, those raised in a constant dark condition showed reduced γH2AV accumulation and nuclear expansion (Fig. 3h). In addition, expression of human superoxide dismutase (hSOD1), a scavenger of reactive oxygen species, in $Rh1 > TER94^{K2A}$ background reduced γH2AV levels (Fig. 3i). The results from these two independent regimens of mutagen exposure minimization suggest that the increased γH2AV in $Rh1 > TER94^{K2A}$ is a consequence of accumulating unrepaired DNA damages from environmental genotoxic agents.

**Accumulation of *Drosophila* MDC1 phenocopies TER94^{K2A}-associated nuclear enlargement.** The increase of DDR markers, as well as FK2 accumulation (Fig. 2), in $Rh1 > TER94^{K2A}$ suggests that, in the absence of TER94 function, one or a few DDR proteins accumulated, leading to enlarged nuclei. To test this hypothesis, we asked whether reduction of certain DDR proteins, which might be TER94 substrates, could suppress the nucleus expansion in $Rh1 > TER94^{K2A}$. Conversely, if accumulation of certain DDR proteins is responsible for the nuclear size increase in $Rh1 > TER94^{K2A}$, direct overexpression of these DDR proteins should recapitulate the enlarged nuclei phenotype. Using these two criteria, we screened DDR genes by RNA interference (RNAi) and $Rh1$-$GAL4$-driven overexpression. RNAi-mediated knockdowns of several candidates, including *tefu*, *l(3)mbt*, *mu2*, *bon*, *dgrn*, and *p53* (Fig. 4a and Supplementary Fig. 9a) showed suppression of the nuclear enlargement, although only $Rh1 > eGFP$-*Mu2*, among the tested constructs, presented a nuclear expansion phenotype similar to those seen in TER94^{K2A} (Fig. 4b, c and Supplementary Fig. 9b). We made $UAS$-*Mu2* transgenic flies and showed $Rh1 > Mu2$ photoreceptors contained enlarged nuclei (Fig. 5a), verifying that this *Mu2*-overexpression phenotype was unaffected by the GFP moiety. $Rh1 > eGFP$-*Mu2*, similar to $Rh1 > TER94^{K2A}$, showed increased γH2AV staining (Fig. 4f, g), consistent with our notion that accumulation of DDR proteins hinders DNA damage repair. Moreover, the eGFP-Mu2 signal intensity in $Rh1 > eGFP$-*Mu2* correlated positively with the nuclear size increase (Fig. 4h), strengthening our hypothesis that the inability to remove Mu2 from the chromatin in TER94^{K2A} results in nuclear expansion. Co-expression of Mu2 in cells deficient in TER94 function resulted in strong nuclear FK2 signals (Supplementary Fig. 10), suggesting Mu2 is one of the ubiquitinated proteins accumulated in $Rh1 > TER94^{K2A}$. However, $Rh1 > Mu2$ by itself did not increase FK2 staining in the nucleus (Supplementary Fig. 10), suggesting that accumulation of ubiquitinated proteins per se is not a cause of nuclear enlargement.

The similarities between TER94^{K2A} and Mu2-overexpression phenotypes suggest that these two genes act on the same pathway.

Indeed, several lines of evidence suggest that Mu2 is a substrate of TER94. Co-overexpression of wild-type TER94 reduced eGFP-Mu2 signals in the nuclei, and conversely, reduction of TER94 function by RNAi increased nuclear eGFP-Mu2 signals (Fig. 4b, d, i). This influence of TER94 on Mu2 level was functional, as TER94^{WT} overexpression suppressed and TER94 knockdown enhanced Mu2-induced nuclear enlargement, respectively (Fig. 4b, c, e). It is worth noting that, as both TER94 and Mu2 were expressed from a heterologous promoter (from UAS directed by $Rh1$-$GAL4$), this effect of TER94 on apparent Mu2 expression is unlikely to be transcriptional. Indeed, RT-PCR showed the level of Mu2 transcripts was unaffected in TER94^{K2A}-expressing cells (Fig. 4j). Nevertheless, Western blot of TER94^{K2A} lysates showed an increase in the endogenous Mu2 (Fig. 4k), indicating that loss of TER94 function increased Mu2 protein level via a post-transcriptional mechanism. The in situ proximity ligation assay (PLA) showed that TER94 interact with Mu2 in vivo (Fig. 4l). The fact that TER94 regulates Mu2 abundance, along with this observation that these two proteins are in close proximity, suggests that Mu2 is a potential TER94 substrate.

**p53A acts downstream of MDC1 in TER94 dysfunction-induced nuclear expansion.** To identify downstream effectors that contribute to this nuclear size increase, we looked for proteins that are known to interact with Mu2 in literature and have significantly modified the TER94^{K2A}-caused nuclear expansion in the abovementioned RNAi screen. One such candidate is p53, which is known to interact with MDC1 in mammals[47,48]. In addition, RNAi-mediated downregulation of p53 suppressed both TER94^{K2A} and Mu2-induced nuclear expansion (Figs. 4a and 5a). To independently confirm this role of p53 in nuclear enlargement, we overexpressed TER94^{K2A} or Mu2 in homozygous p53 null ($p53^{5A-1-4}$; labeled as $p53^{-/-}$)[49] background. Similar to the knockdown, a complete removal of p53 function suppressed TER94^{K2A}- and Mu2-associated nuclear expansion (Fig. 5b, c). These genetic interactions place p53 function downstream of TER94 and Mu2 in this context.

The *Drosophila* p53 gene encodes four isoforms, A/C, B, D, and E (the isoform C the same as isoform A), structurally differing by the length of the N-terminal transactivation domain (TAD)[50]. While the abovementioned genetic interactions link p53 to the TER94^{K2A}-associated nuclear size increase, it is unclear whether this particular p53 role is isoform-specific. Quantitative Western analysis of $Rh1 > TER94^{K2A}$ adult head extract detected a significant increase of p53A (Fig. 5d). To verify which p53 isoform(s) facilitates this TER94^{K2A}-dependent nuclear expansion, we took advantage of the observation that $p53^{-/-}$

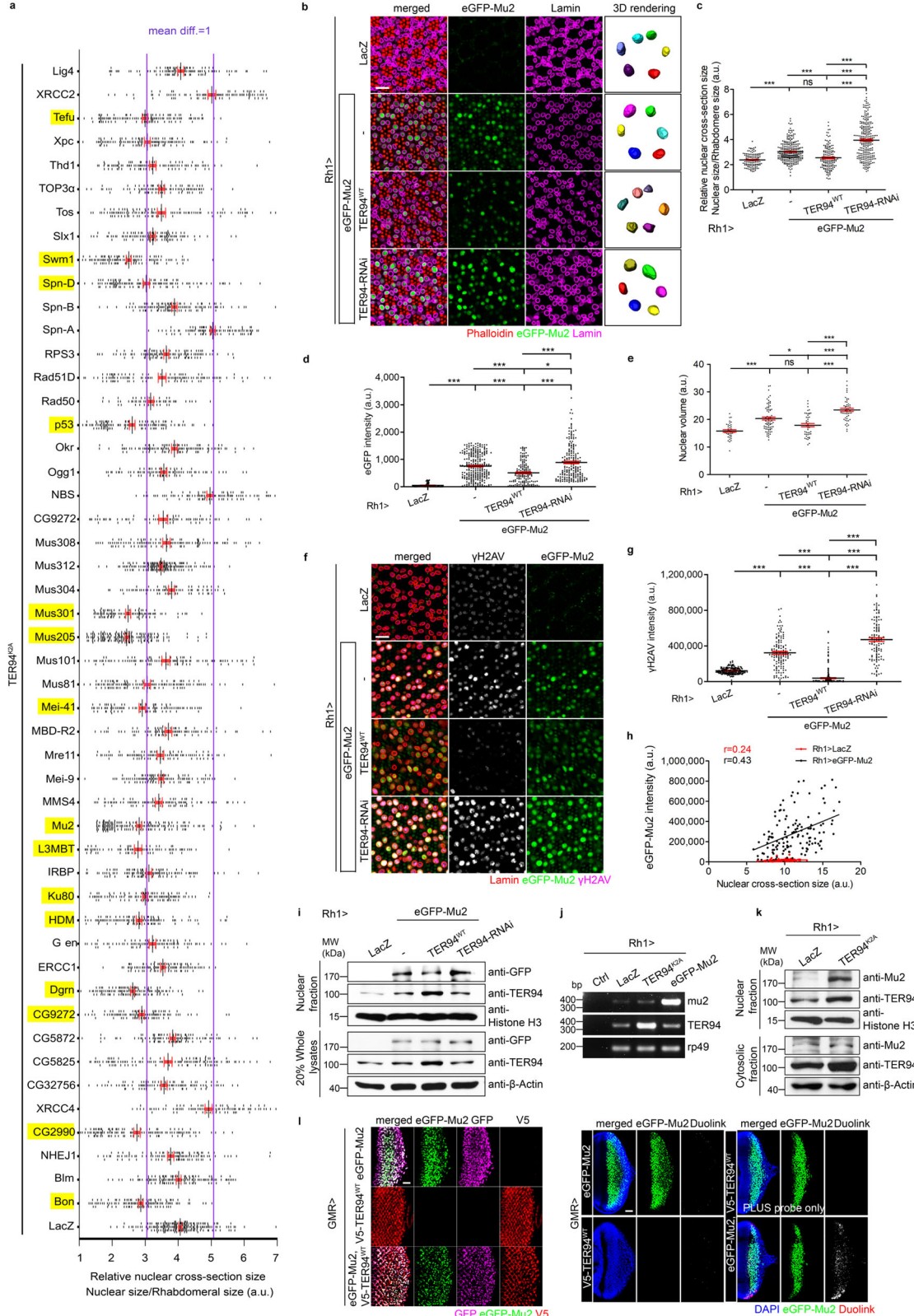

suppressed the Mu2-associated nuclear enlargement (see above). In $Rh1 > eGFP\text{-}Mu2$; p53$^{-/-}$ retina, expression of p53B with $Rh1\text{-}GAL4$ driver had no effect on the nuclear size (Fig. 5c), indicating that this isoform has little, if any, role in this context. In contrast, expression of p53A in the same background restored the nuclear size increase (Fig. 5c), strongly suggesting that this isoform

contributes to the TER94$^{K2A}$-linked nuclear expansion phenotype.

In mammals, p53 proteins are labile[51], and can be stabilized by MDC1 under genotoxic stress[52]. p53 interacts with MDC1 directly through the BRCT (BRCA1 C-terminal) domain[47,48], which is conserved in *Drosophila* Mu2[53]. Thus we hypothesized

**Fig. 4 TER94 influences Mu2 level in regulating nuclear size. a** Quantification of the R1-R6 cross-section nuclear area from $Rh1 > TER94^{K2A}$ flies co-expressing *UAS-RNAi* constructs targeting indicated genes. The purple lines indicate the difference of the mean equals 1 (mean diff. = 1) as compared to the control ($Rh1 > TER94^{K2A}$, *LacZ*). The DDR genes with mean diff. ≥1 are marked in yellow. The number of independent nuclei measured are listed in Statistics and reproducibility. **b** Confocal and 3D rendering images (individual nuclei are color-coded) of 7-day-old (confocal) or 5-day-old (3D rendering) retinas from flies expressing indicated transgenes stained with phalloidin (red) and anti-Lamin (magenta). **c–e** Quantification of the cross-section area of R1-R6 nuclei (**c**), eGFP intensity (**d**), and nuclear volume (**e**). **f** Confocal images of 7-day-old retinas expressing indicated transgenes stained with anti-Lamin (red) and anti-γH2AV (magenta). **g** Quantification of the anti-γH2AV intensity. **h** Pearson's correlation of nuclear size and eGFP-Mu2 intensity (r: coefficient). **i** Western analysis of eGFP-Mu2 level (anti-GFP) in indicated TER94 backgrounds. Anti-Histone H3 and anti-β-Actin serve as loading controls. **j** RT-PCR measurement of *mu2* and *TER94* expression from indicated genotypes. The control (Ctrl) groups omitted templates. rp49 serves as control. **k** Western analysis of Mu2 level in tissues expressing LacZ or TER94$^{K2A}$. Anti-Histone H3 and anti-β-Actin serve as loading controls. **l** Larval eye discs from flies expressing indicated transgenes under the control of *GMR-GAL4* are stained with anti-GFP (magenta) and anti-V5 (red) (left panel). Larvae bearing the same genotypes are subjected to in situ PLA using anti-GFP and anti-V5 antibodies (right panel). The duolink signal reveals proteins in the same complex. The lines in dot plots from **a** represent mean ± SE. The number of independent experiments performed: 3 (**b**, **i**), 2 (**j**, **k**, **l**). Quantification details of **c**, **d**, **e**, and **g** are listed in Statistics and reproducibility. Scale bars: 10 μm (**b**, **f**), 20 μm (**l**).

that Mu2 binds with and stabilizes p53 in affecting nuclear size. In support of this, endogenous p53 was increased in $Rh1 > TER94^{K2A}$ (Fig. 5d), and exogenously expressed p53A was elevated in tissues co-expressing $TER94^{K2A}$ (Fig. 5e, g). Similarly, ectopically expressed p53A was upregulated in Mu2-overexpression and reduced in Mu2-RNAi backgrounds, respectively (Fig. 5e). In situ PLA demonstrated that tagged Mu2 could interact with p53A in eye discs (Fig. 5f). Taken together, these data support a cascade, in which TER94$^{K2A}$ increases Mu2 abundance to stabilize p53A. In addition, the stabilized p53 might be an effector mediating the nuclear expansion.

To ask whether stabilized p53A can directly influence nuclear size, we overexpressed Myc-tagged p53A with *Rh1-GAL4*. Although *Rh1-GAL4* is active in all outer R cells, only a small subset of them exhibited detectable levels of Myc-p53A (Fig. 5g), consistent with the notion that p53 proteins are labile. Still, those with sufficient Myc-p53A accumulation exhibited enlarged nuclei, demonstrating that p53A expression alone can increase nuclear size. Importantly, $Rh1 > Myc$-p53A showed no increase in γH2AV (Supplementary Fig. 11), suggesting that accumulation of DNA lesions and nuclear enlargement can be uncoupled.

**TER94$^{K2A}$ nuclear expansion is associated with p53-mediated autophagy flux impairment.** Expression of *Drosophila* p53A in photoreceptors has been reported to impair autophagic flux[54]. As p53A contributes specifically to the nuclear enlargement, we hypothesized that the Atg8a and p62 accumulation observed in TER94$^{K2A}$ (Fig. 2d, e) might represent arrested autophagosomes. In this scenario, disrupted autophagic flux could cause the nuclear size increase in $Rh1 > TER94^{K2A}$ by failing to remove damaged nuclear components. To ask whether this autophagic disruption in $Rh1 > TER94^{K2A}$ depends on Mu2 or p53, we monitored the signal intensities of GFP-Atg8a or GFP-p62 in $Rh1 > TER94^{K2A}$ eyes co-expressing RNAi constructs against p53 or Mu2. Compared to $Rh1 > TER94^{K2A}$ alone, knockdowns of p53 and Mu2 significantly reduced the GFP-Atg8a and GFP-p62 signals (Fig. 6a–c). Antibodies staining for endogenous Atg8a and p62 levels confirmed this dependence of $Rh1 > TER94^{K2A}$ on p53 or Mu2 (Supplementary Fig. 12), demonstrating that TER94$^{K2A}$ perturbs autophagy through a Mu2- and p53-dependent manner.

To further understand the nature of this autophagic disruption in $Rh1 > TER94^{K2A}$, we used a tandem GFP-mCherry-Atg8a fluorescent reporter, which allows us to determine, by a ratiometric analysis of GFP and mCherry fluorescence, the state of autophagy progression. As the GFP moiety in the chimera is quenched by the acidic environment during autophagosome–lysosome fusion, this reporter in $Rh1 > LacZ$ control exhibited a signature puncta proportion in various stages: ~58% in stage I (autophagosomal formation; yellow), ~6% in

stage II (fusion with lysosomes; red), and ~30% in stage III (lysosomal degradation; small red). Treating the flies with rapamycin, which induces autophagy, decreased the proportion of puncta in stage I to ~13%, with a concomitant increase of stage II puncta to ~62%. In contrast, treatment with chloroquine, known to inhibit lysosomal fusion[55], increased the proportion of stage I puncta to ~83%. In $Rh1 > TER94^{K2A}$, ~93% of tandem reporter puncta were in stage I. This phenotypic similarity to chloroquine-treated retina (Fig. 6d–g) suggests that TER94 dysfunction causes an arrest of autophagy progression. Notably, we observed diffused signals of tandem reporter (Fig. 6d) in the nuclei, a phenotype also seen with the GFP-Atg8a reporter (Fig. 2d). Because LC3/Atg8a can reside in the nucleoplasm and cytoplasm, we surmised that TER94$^{K2A}$ hinders its exportation from the nucleus, a process promoted by autophagy activation[56,57]. To ensure that this disruption of autophagic flux is specific to this particular TER94 function, we showed that compared to $Rh1 > LacZ$ control (~59%, Fig. 6h–j), the proportion of stage I puncta was also higher (~88%) in $Rh1 > Mu2$.

To confirm the autophagy flux impairment and nuclear size enlargement are facilitated by p53A, we compared the endogenous p62 levels and the nuclear sizes in R cells expressing Mu2 alone with those co-expressing p53A. Compared to LacZ control, Mu2 overexpression alone showed a slight but reproducible increase of p62. The p62 signal in $Rh1 > Mu2$ was strongly enhanced when p53A were co-expressed. Co-expression of p53B showed little effect on p62 signal, demonstrating the specificity of the enhancement by p53A (Fig. 6k). Likewise, the moderately enlarged nuclear size in $Rh1 > Mu2$ expanded further with p53A co-expression (Fig. 6k). Together, these results show that TER94$^{K2A}$ and Mu2 overexpression cause autophagic flux defect and nuclear enlargement through p53A.

## Discussion

We have used the *Drosophila* eye to investigate the physiological function of VCP and the mechanism by which specific VCP alleles contribute to neurodegeneration[16]. Here, using both cell-specific expression of dominant-negative TER94 mutants and FLP-induced $TER94^{k15502}$ mutant clones, we show that removal of TER94 function in postmitotic cells leads to an aberrant nuclear expansion. This nuclear size increase is not restricted to photoreceptors, as the expression of TER94$^{K2A}$ in other cell types generates a similar phenotype, suggesting that the role of TER94 in nuclear size regulation is fundamental to most cells. Altered nuclear size has been associated with diseases like cancer, although the mechanistic link between aberrant nuclear size and cell pathophysiology is not well understood. While multiple explanations likely exist for aberrant nuclear enlargement, the known roles of VCP in degenerative disorders and the specificity

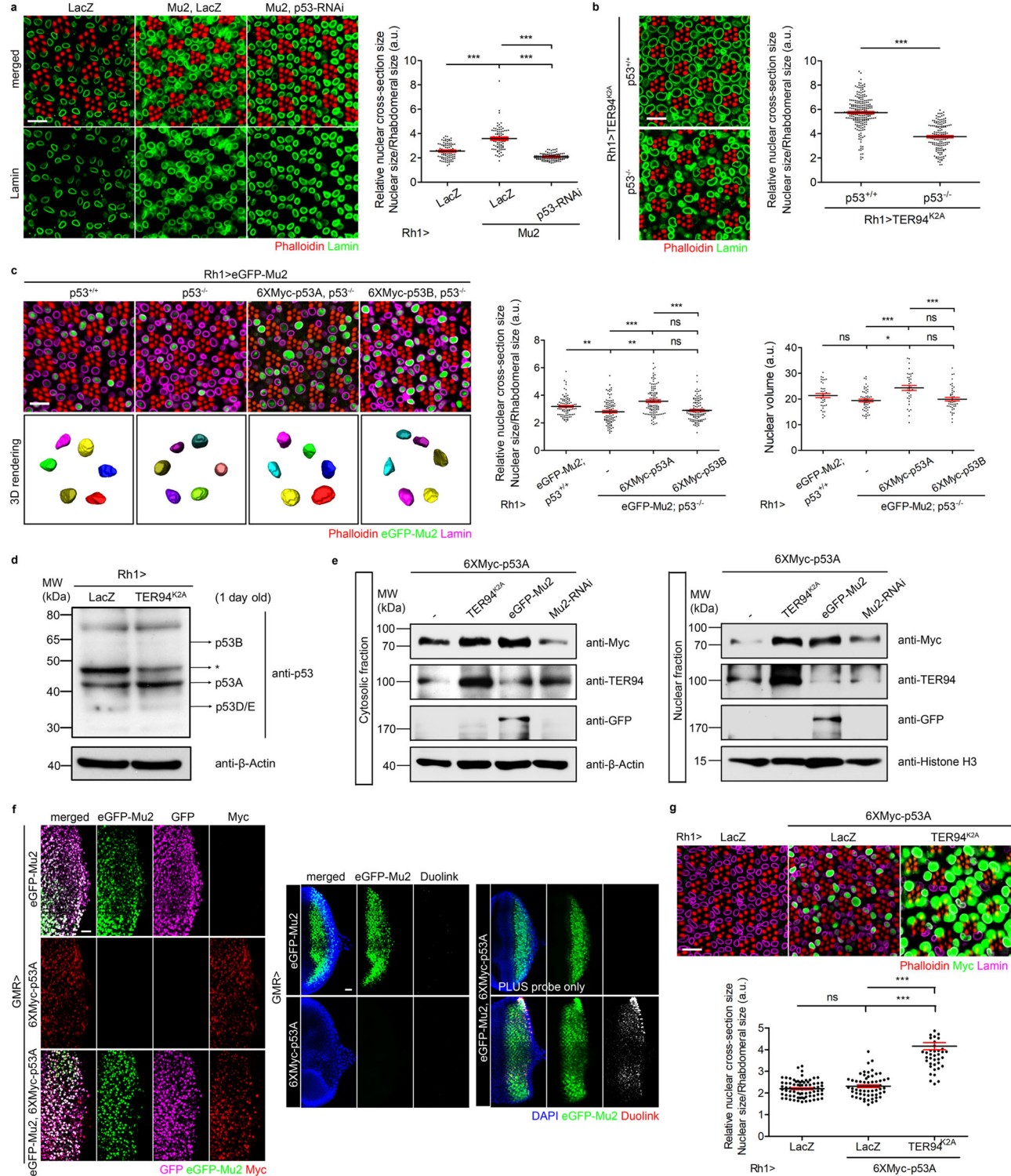

of TER94$^{K2A}$ nuclear enlargement suggest this is an ideal model system to explore the underlying mechanism.

As the molecular function of VCP has been extensively studied, we suspect that the nuclear enlargement is linked to one of its characterized roles. We have previously shown that over-expression of Derlin-1, a direct interactor of TER94, retains TER94 in the cytoplasm[40]. Derlin-1-dependent depletion of nuclear TER94 causes enlarged nuclei, suggesting that this nuclear enlargement is caused by a disruption of TER94 function

in the nucleus. In support of this, TER94$^{K2A}$-expressing R cells show elevated levels of ubiquitinated proteins inside the nucleus, and the level of ubiquitination correlates well with the severity of the nuclear size. Our results suggest that the clearance of some ubiquitinated nuclear proteins is blocked by TER94$^{K2A}$, thereby causing nuclear enlargement. While accumulation of nuclear proteins, caused by a disruption of bulk nucleocytoplasmic transport, has been shown to affect N/C ratio in yeast[15] and mammalian cells[58], our analysis suggests that these accumulated

**Fig. 5 p53 acts downstream of Mu2 in TER94 dysfunction-induced nuclear expansion. a** Left: Confocal micrographs of 5-day-old control (*Rh1 > LacZ*) or *Rh1 > Mu2* adult retinas co-expressing LacZ or p53-RNAi stained with phalloidin (red) and anti-Lamin (magenta). Right: Quantification of the cross-section area of R1-R6 nuclei. **b** Left: Confocal images of 6-day-old *Rh1 > TER94*$^{K2A}$ adult retinas in wild type (+/+) or *p53* null (−/−) background stained with phalloidin (red) and anti-Lamin (green) antibody. Right: Quantification of the cross-section area of R1-R6 nuclei. **c** Left: Confocal and 3D rendering images (individual nuclei are color-coded) of 5-day-old *Rh1 > eGFP-Mu2* adult retinas in wild type (+/+), *p53* null (−/−), 6XMyc-p53A overexpression in *p53* null (−/−), and 6XMyc-p53B overexpression in *p53* null (−/−) backgrounds stained with phalloidin (red) and anti-Lamin (magenta). Right: Quantification of the cross-section area and the volume of R1-R6 nuclei. **d** Western analysis of p53 level in tissues expressing LacZ (control) or TER94$^{K2A}$ under the control of *Rh1-GAL4*. The arrows indicate molecular weight of each p53 isoforms. Asterisk: nonspecific. Two independent experiments showed similar results. **e** Western analysis of the Myc-p53A level in tissues expressing indicated genes. Anti-Histone H3 and anti-β-Actin serve as loading controls. **f** Larval eye discs expressing indicated genes are stained with anti-GFP (magenta) and anti-Myc (red) antibodies (left panel). Larvae bearing the same genotypes are subjected to in situ PLA using anti-GFP and anti-Myc antibodies (right panel). The duolink signal reveals proteins in the same complex. **g** Top: Confocal micrographs of 5-day-old *Rh1 > LacZ, Rh1 > 6XMyc-p53A > LacZ, and Rh1 > 6XMyc-p53A > TER94*$^{K2A}$ retinas stained with phalloidin (red), anti-Lamin (magenta), and anti-Myc (green). Bottom: Quantification of the cross-section area of R1-R6 nuclei. The number of independent experiments performed: 3 (**b**), 2 (**f**). Quantification details of **a**, **b**, **c**, and **g** are listed in Statistics and reproducibility. Scale bars: 10 μm (**a–c, g**), 20 μm (**f**).

nuclear proteins in *Rh1 > TER94*$^{K2A}$ are limited to DDR factors.

Within the nucleus, VCP has been shown to rapidly accumulate at DNA damage site, facilitating the extraction of chromatin-tethering proteins such as L3MBTL1 and KAP1 to assist the recruitment of 53BP1 or BRCA1, respectively[59,60], and the removal of Ku80 from DNA at the end of non-homologous end joining repair[61]. Knockdowns of fly L3MBT and Ku80 suppress the *Rh1 > TER94*$^{K2A}$-associated nuclear enlargement, linking DNA repair to this aberrant nuclear size. This hypothesis is further supported by our analysis of MDC1, a scaffold protein that bridges the injured DNA and DDR factors[52,62,63]. Mu2, the fly homolog of MDC1, modifies the phenotype of *Rh1 > TER94*$^{K2A}$, and Mu2 overexpression recapitulates the phenotype of *Rh1 > TER94*$^{K2A}$, placing Mu2 downstream of TER94 in this context. As TER94 is pleiotropic, this evidence of Mu2 overexpression phenocopies TER94$^{K2A}$ supports that DDR participates specifically in nuclear expansion. MDC1 is indispensable for DDR; however, prolonged retention of MDC1 on DNA damage site impedes the access of downstream signaling or repairing proteins[28,29,64]. In support of the notion that a coordinated removal of MDC1-γH2AX association is critical for maintaining genomic stability, we observed persistent γH2AV signals in Mu2-overexpressing cells. Interestingly, Mu2 overexpression enlarges the nuclei without increasing FK2 staining, suggesting it is the elevated Mu2 level inside the nucleus, not the accumulation of ubiquitinated nuclear proteins per se, that causes TER94$^{K2A}$-dependent nuclear expansion. In addition, as the SUMO-targeted ubiquitin ligase RNF4 and the deubiquitinase Ataxin-3 have been shown to regulate the removal of MDC1 from chromatin[65], it is likely that the ubiquitinated nuclear proteins observed in *Rh1 > TER94*$^{K2A}$ include Mu2. Consistent with this, we demonstrate that TER94 negatively influences Mu2 level via a post-transcriptional manner, and co-expression of Mu2 significantly increases FK2 signals in *Rh1 > TER94*$^{K2A}$. These observations, combined with the result of in situ PLA demonstrating that TER94 interacts with Mu2, imply that Mu2 is a direct substrate of TER94. Thus, we propose that TER94-mediated degradation of Mu2 is a crucial step permitting the DNA repair machinery to engage the lesion site.

During DDR, MDC1 serves as a binding platform for recruiting downstream DDR factors[52,62] including p53. Our observation that p53-RNAi suppresses both TER94$^{K2A}$ and Mu2-induced nucleus expansion places p53 downstream of these two proteins in promoting nuclear size increase. The PLA assay confirms an interaction between Mu2 and p53. Furthermore, the Western blotting results show Mu2 overexpression boosts p53 levels, suggesting that this Mu2–p53 interaction stabilizes labile p53 proteins. In mammalian systems, p53 protein level under normal condition is kept low, as most of p53 proteins are destined

for degradation via a continuous ubiquitination by E3 ubiquitin ligase MDM2. In the presence of DNA damage, phosphorylation of p53 by various kinases could disturb this MDM2-p53 interaction and stabilize p53[66]. It has been reported that MDC1 is indispensable for CHK2-mediated p53 phosphorylation and stabilization in response to ionizing radiation in mammalian cells[52]. The observation that Mu2 overexpression positively and Mu2-RNAi negatively regulates p53 protein level, respectively, suggests a similar mechanism may exist in *Drosophila*. In support of this, MDC1 is known to interact with p53 via the BRCT domain[47,48], which is conserved in *Drosophila* Mu2[53]. It is thought that the accumulated Mu2 represents a signal from stalled DDR, and this Mu2–p53 interaction allows p53 to elicit responses to protect the genome. Our demonstration that manipulating p53 level modifies nuclear size under TER94$^{K2A}$ or Mu2 overexpression suggests that alteration in nucleus structural maintenance could be a response to stalled DDR.

LINC complexes, implicated in N/C control[14], comprise inner and outer nuclear membrane proteins and function to bridge the nuclear lamina and the cytoskeleton[67]. In yeast and mammalian cells, chromatin associated with DSBs exhibited increased mobility and this phenomenon requires LINC complexes[68], raising a potential link between chromatin repair to nuclear architecture. Our analysis suggests p53 may participate in this or a similar process, possibly functioning as a sensor of unrepaired DNA damages for machinery that regulates the nuclear size. Indeed, 53BP1, a p53 binding protein, decorates the damaged foci[68], although whether p53 has a direct role in this process remains to be shown.

The p53 locus is capable of generating multiple isoforms through the usage of alternative promoters, splicing sites, and translational initiation sites[69]. Our study suggests the nuclear expansion, caused by TER94 dysfunction or Mu2-overexpression, selectively requires p53A. Importantly, expression of p53A alone can increase nuclear size in the absence of excessive DNA damages, placing its role downstream of unrepaired DNA damages in TER94$^{K2A}$-dependent nuclear expansion. A recent report suggests that *Drosophila* p53A and p53B have opposing effect on autophagy and apoptosis, and it is the balance between these p53 isoforms that mediates this cellular life-and-death decision under stress[54]. Our observation that p53A is selectively required for this context suggests that deregulated DDR, caused by either TER94 dysfunction or Mu2 accumulation, tips the balance to favor p53A-dependent response. In any case, our work provides a new direction in resolving p53-regulated cellular events and emphasizes the importance of unveiling factors controlling p53 abundance in an isoform-specific fashion.

*Drosophila* p53A have recently been reported to disrupt autophagy flux[54]. As our analysis implicates p53A, it seems plausible that a disruption of autophagy has a role in this

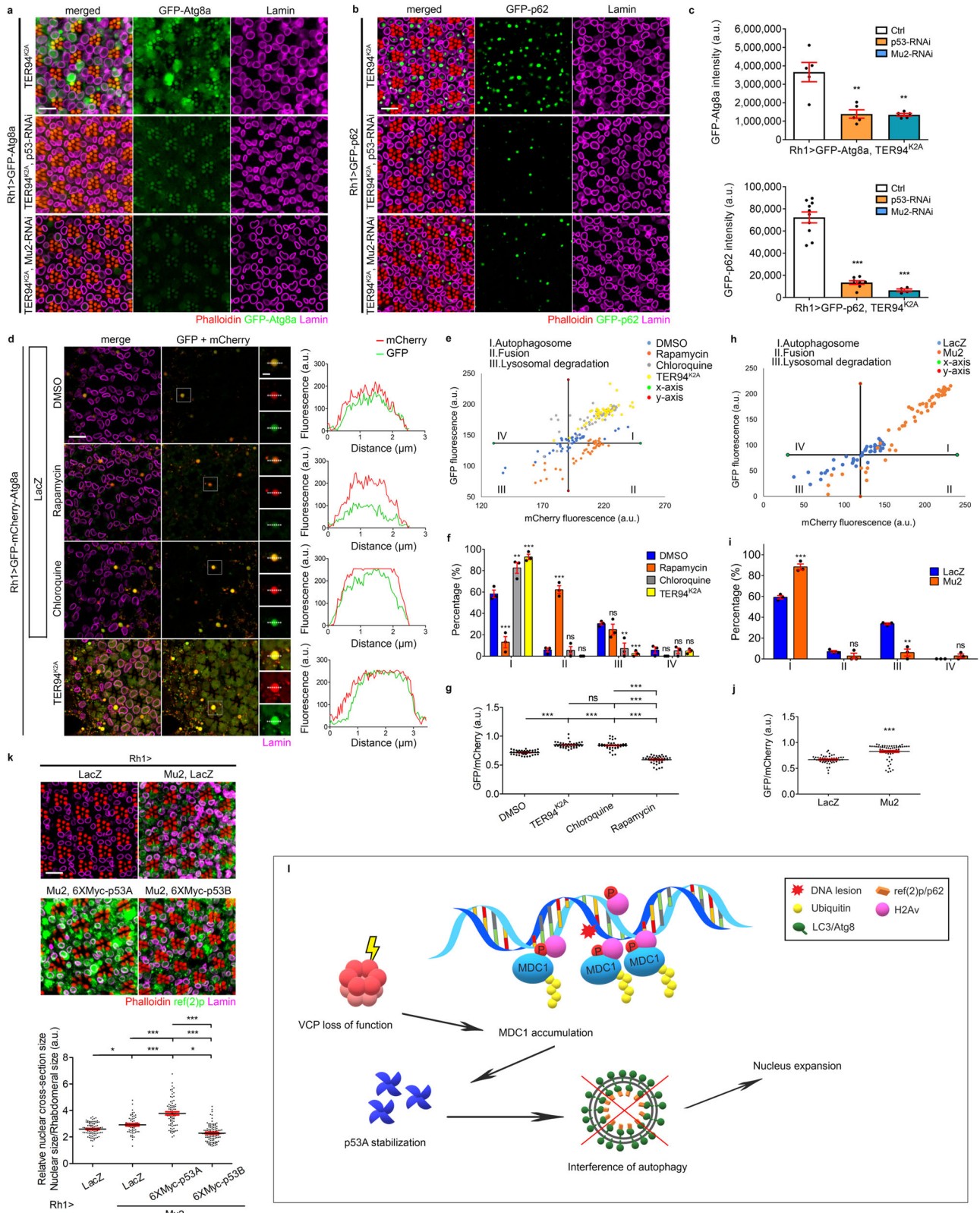

TER94[K2A]-associated nuclear enlargement. Indeed, TER94[K2A] cells exhibit age-dependent p62 accumulation, a well-established indicator of autophagy deficiency[39]. Furthermore, downregulation of Mu2 or p53 ameliorates the p62 accumulation in *Rh1 > TER94[K2A]*, indicating that this autophagy deficiency is mediated by Mu2 and p53. In addition to regulating chromatin

modification[70], autophagy has been shown to remove nuclear components upon genotoxic insults[71]. Thus, p53A expression may disrupt autophagy-mediated removal of nuclear components, thereby increasing the nuclear size. Taken together, we propose a model (Fig. 6l) that VCP dysfunction causes aberrant accumulation of MDC1 on the DNA damage foci. The MDC1

**Fig. 6 TER94 dysfunction triggers nuclear expansion through p53A-mediated autophagy blockage. a, b** Confocal micrographs of 7-day-old *Rh1 >TER94^K2A > GFP-Atg8a* (**a**) or *Rh1 > TER94^K2A > GFP-p62* (**b**) retinas co-expressing p53-RNAi or Mu2-RNAi stained with phalloidin (red) and anti-Lamin (magenta). **c** Quantification of the GFP-Atg8a (upper panel) or the GFP-p62 (lower panel) intensity. In total, ≥4 independent eyes are measured. **d** Left: Confocal micrographs of 5-day-old retinas from *Rh1 > LacZ* or *Rh1 > TER94^K2A* co-expressing GFP-mCherry-Atg8a. Control flies are treated by the indicated drugs. Anti-Lamin (magenta) marks nuclei. Puncta shown in separated channels are enlarged from the corresponded white boxes. Right: Fluorescence intensity plots of GFP and mCherry across the corresponded puncta. **e** Dot plot of mCherry and GFP fluorescent intensity from the tested group in **d**. The determination of *X*- and *Y*-axis and the definition of each quadrant are disclosed in Methods. **f** Quantification of puncta distribution in each quadrant. **g** Ratiometric analysis of GFP/mCherry fluorescent intensity. Puncta from 6 independent eyes are measured. **h** Dot plot of mCherry and GFP intensity from puncta in *Rh1 > LacZ* and *Rh1 > Mu2* flies co-expressing GFP-mCherry-Atg8a. **i** Quantification of puncta distribution in each quadrant and **j** ratiometric analysis of GFP/mCherry fluorescent intensity. Puncta from 6 independent eyes are measured. **k** Top: Confocal micrographs of 5-day-old adult retinas expressing LacZ, and Mu2 with LacZ or indicated p53 isoforms stained with phalloidin (red), anti-Lamin (magenta), and anti-ref(2)p (green). Bottom: Quantification of the cross-section area of R1-R6 nuclei. Quantification details of **c**, **f**, **g**, **i**, **j**, and **k** are listed in Statistics and reproducibility. Scale bars: 10 µm (**a**, **b**, **d** regular panels, **k**), 2 µm (**d** enlarged panels). **l** A model depicts loss of VCP function triggers MDC1–p53-mediated autophagy blockage and nuclear expansion. VCP regulates MDC1 retention on damaged foci. Loss of VCP function causes MDC1 accumulation by which to stabilize its interactor p53A, which interferes autophagy and causes nuclear expansion.

accumulation then stabilizes p53A, which hinders the autophagy-dependent removal of nuclear materials and alters the nuclear size.

Nearly 150 years ago, Sir Lionel Beale, the father of cytology, described different nuclear morphology in various tumor types and recognized its prognostic value[72]. Nuclear morphometry, including changes in nuclear size or shape, and the presence of nuclear lobulation, have since been extensively used to grade different tumors and optimize clinical treatment. As altered VCP expression is also associated with poor prognosis and sometimes linked to the metastatic potential in several cancer types[73–78], our study provides a plausible explanation of how VCP dysfunction, the nuclear size change, and defective DDR could be linked. Furthermore, the identification of MDC1 is a key player in stabilizing p53 during DDR, and by which to modulate autophagy and nuclear size opens a new direction for the field studying p53-mediated cellular responses. Our findings might help in developing new targets for clinical applications.

## Methods

***Drosophila* genetics and molecular biology.** Flies were raised on standard cornmeal food at 25 °C under 12 hours light/dark cycles unless otherwise mentioned. The following *Drosophila* strains were used: Caton-S (CS), UAS-LacZ, *Rh4-GAL4, Spa-GAL4, Mhc-GAL4, UAS-tub-GAL80^ts, UAS-GFP-NLS, hs-FLP, FRT42D GMR-myr.GFP, UAS-CD4-mCherry, UAS-EGFP.Mu2*, Fly-FUCCI (*UAS-GFP-E2F1_{1-230}, UAS-mRFP1-NLS-CycB_{1-266}*), p53^5A-1-4, UAS-GFP-mCherry-Atg8a (Bloomington *Drosophila* Stock Center, Indiana, USA), *FRT42D TER94^K15502, UAS-S/G2/M-Green* (Kyoto Stock Center, Kyoto, Japan), *GMR-GAL4* (originally derived from Dr. Matthew Freeman[79]), *Rh1-GAL4* (provided by Dr. Larry Zipursky), *UAS-CD8-PARP-Venus* (gift of Dr. Darren Williams), *UAS-GFP-Atg8a*, and *UAS-GFP-p62* (originally derived from Dr. Thomas Neufeld), *UAS-6XMyc-p53* isoforms (kindly provided by Dr. Brian Calvi). All of the transgenic RNAi lines were sourced from the *Drosophila* Stock Center (Bloomington, Indiana, USA) and the Vienna *Drosophila* RNAi Center (Vienna, Austria). *UAS-CD3δ-YFP, UAS-TER94^WT, UAS-TER94^A229E, UAS-TER94^K2A, UAS-TER94^E2Q, UAS-Derlin-1*, and *UAS-Derlin-1^{ΔSHP}* have been previously described[16,40].

To generate somatic eye clones of *TER94^K15502*, *hs-FLP; FRT42D, GMR-myr. GFP* virgins were crossed with *FRT42D, TER94^{K15502}/ CyO* males. The F1 progeny were heat-shocked in a 37 °C water bath for 1 h at second or third instar larval stage to induce mitotic recombination. An additional 1-hour heat-shock a 37 °C on the following day was administered to increase the frequency of recombination. Eye clones were examined among F1 adults without CyO.

To generate *pUAST-Mu2* construct, *Drosophila* Mu2 was excised from cDNA clone LD44171 (*Drosophila* Genomics Resource Center, Indiana, USA) and subcloned into *pUAST* as a KpnI-XbaI fragment. The *pUAST-Mu2* construct was verified by sequencing prior to transgenic fly production by P element-mediated transformation[80] (WellGenetics, Taiwan).

**Antibody production and immunohistochemistry.** To generate polyclonal anti-TER94, anti-p53, and anti-Mu2 antibodies, polypeptides corresponding to the amino acids 701-801 of TER94, amino acids 373-495 of p53, and the amino acids 627-947 of Mu2, respectively, were bacterially expressed, purified, and used for immunization (GeneTex). Whole-mount preparation of fly eyes and eye discs were

performed as previously described[81,82]. Primary antibodies used were the following: mouse anti-Lamin Dm0 (1:20, ADL67.10, Developmental Studies Hybridoma Bank, DSHB), rabbit anti-Lamin Dm0 (1:2,000, a generous gift from Dr. Paul Fisher), customized rabbit anti-TER94 (1:500), mouse anti-Nuclear Pore Complex Proteins (NPC) (1:100, ab24609, Abcam), mouse anti- mono- and poly-ubiquitinated conjugates (FK2) (1:100, BLM-PW8810-0100, Enzo Life Sciences), rabbit anti-ref(2)p (1:20, ab178440, Abcam), rabbit anti-LC3A/B (1:20, ab128025, Abcam), rabbit anti-V5 (1:200, AB3792, Millipore), rabbit anti-Atg8 (1:100, ABC974, Millipore), mouse anti-γH2Av (1:200, UNC93-5.2.1, DSHB), mouse anti-PCNA (1:20, ab29, Abcam), mouse anti-GFP (1:500, 4C9, DSHB), mouse anti-Myc (1:100, GTX75953, GeneTex), rabbit anti-mCherry (1:100, GTX128508, GeneTex), rabbit anti-cleaved PARP (1:50, ab2317, Abcam), rabbit anti-cleaved Caspase 3 (1:20, 9661 s, Cell Signaling). Alexa Fluor® 488, Alexa Fluor® 647, Cy3, and Cy5 conjugated secondary antibodies: Alexa Fluor® 488 AffiniPure Goat Anti-Mouse IgG (H + L) (115-545-146, Jackson ImmunoResearch Laboratories), Alexa Fluor® 488 AffiniPure Goat Anti-Rabbit IgG (H + L) (111-545-144, Jackson ImmunoResearch Laboratories), Alexa Fluor® 647 AffiniPure Goat Anti-Mouse IgG (H + L) (115-605-146, Jackson ImmunoResearch Laboratories), Cy™3 AffiniPure Goat Anti-Mouse IgG (H + L) (115-165-146, Jackson ImmunoResearch Laboratories), Cy™5 AffiniPure Goat Anti-Mouse IgG (H + L) (115-175-146, Jackson ImmunoResearch Laboratories), Cy™3 AffiniPure Goat Anti-Rabbit IgG (H + L) (111-165-144, Jackson ImmunoResearch Laboratories), Cy™5 AffiniPure Goat Anti-Rabbit IgG (H + L) (111-175-144, Jackson ImmunoResearch Laboratories) were used at 1:100 dilutions. F-Actin enriched rhabdomere was labeled by Rhodamine-conjugated phalloidin (1:20, P1951, Sigma-Aldrich). Zeiss LSM-510 or 800 confocal microscopes were used for collecting all fluorescent images. Adobe Photoshop 2021 was used for figure preparation. For experiments that comparing fluorescent-labeled probes among different genotypes, the sample preparation and image processing was performed with the same procedure and setting.

**Quantification of the nuclear size.** For all the quantifications of cross-section areas, multiple R cell nuclei, each represented by a dot in figure panels, from at least 2 independent eyes of each group were analyzed with Fiji, an image processing package based on Image J 1.53e. The nuclear and the rhabdomeral areas, represented by Lamin and phalloidin staining, respectively, are manually outlined, and the area of each nucleus is normalized to the area of its corresponding rhabdomere as a relative nuclear cross-section size. For the measurement of nuclear volume, Z-stacks of continuous cross-section images of 1 µm thickness that cover the anti-Lamin signals were collected. Raw data were further processed with Avizo 9.4 (Thermo Fisher Scientific) for nucleus volumetric segmentation and volume/morphology quantification. We first use magic wand tool to segment nucleus in 3D, and further use Material Statistic function for each segmented nucleus volume quantification. To quantify the morphology changes, we use Label Analysis function and define a new parameter "sphericity" $\Psi = \frac{\pi^{\frac{1}{3}}(6V_p)^{\frac{2}{3}}}{A_p}$, where $V_p$ is the volume of each segmented nucleus and $A_p$ is the surface area of the same nucleus. Sphericity can represent if the 3D shape of the nucleus close to a perfect sphere (sphericity of a perfect sphere is 1). The formula we used for Pearson's correlation analysis of nuclear cross-section size and FK2 intensity/eGFP-Mu2 intensity is $r = \frac{n(\sum xy) - (\sum x)(\sum y)}{\sqrt{[n\sum x^2 - (\sum x)^2]}\sqrt{[n\sum y^2 - (\sum y)^2]}}$, in which $r$ is correlation coefficient, $x$ is the nuclear cross-section size, $y$ is the FK2 intensity or eGFP-Mu2 intensity, and $n$ is the number of paired measurement.

**Subcellular fractionation.** Subcellular fractionations to isolate nuclear and cytosolic proteins were performed as described by Nabbi and Riabowol[83] with modifications. Briefly, fly heads of appropriate genotypes were homogenized in ice-cold

PBS containing 0.1% NP-40 supplemented with protease inhibitor cocktail (Roche). While 25% of the homogenates (the whole cell lysates) were saved on ice, the remaining homogenates were centrifuged for 30 s at 9300 x $g$. The resulting supernatant (the cytosolic fraction) was kept on ice, whereas the pellet was resuspended in ice-cold PBS containing 0.1% NP-40 with protease inhibitor cocktail. The resuspended pellet was re-centrifuged for 30 s at 9300  x $g$ and the resulting pellet was resuspended in ice-cold PBS containing 0.1% NP-40 with protease inhibitor cocktail (the nuclear fraction). The whole cell lysates and the nuclear fraction were sonicated on ice with two 8-s pulses at 20 kHz, followed by a centrifugation for 1 min at 16,100 x $g$ to remove the debris. The final whole cell lysates and the nuclear fractions were obtained by collecting the supernatants after centrifugation.

**Immunoblotting.** For Westerns, primary antibodies were used as the following dilutions: customized rabbit anti-TER94 (1:5,000), rabbit anti-Histone H3 (1:40,000, ab1791, Abcam), mouse anti-β-actin (1:20,000, GTX629630, GeneTex), rabbit anti-GFP (1:1,000, GTX113617, GeneTex), mouse anti-Myc (1:5,000, GTX75953, GeneTex), customized rabbit anti-Mu2 (1:1,000), and customized rabbit anti-p53 (1:500). Secondary antibodies conjugated with HRP: Goat anti-Rabbit IgG (HRP) (GTX213110-01, GeneTex) and Goat anti-Mouse IgG (HRP) (GTX213111-01, GeneTex) were used in 1:10,000 dilutions. All loading controls were prepared by stripping off the reagents from the original membrane and then re-immunoblotting with anti-β-actin (for whole cell lysate and cytosolic fraction) or anti-Histone H3 (for nuclear fraction) following the standard procedures.

**RT-PCR.** The total RNA from fly heads of $Rh1 > LacZ$, $Rh1 > TER94^{K2A}$, and $Rh1 > eGFP-Mu2$ were isolated using GENEzol™ TriRNA Pure Kit (Geneaid) following the manufacturer's instructions. Four micrograms of RNA was used for reverse transcription (Super-Script II, Invitrogen) following the manufacturer's instructions. Subsequent PCR amplification was performed with about 1 µg cDNA. Specific primer pairs for TER94, Mu2, and internal control rp49 were listed in Supplementary Table 1.

**In situ protein–protein interaction detection.** The Duolink® Proximity Ligation Assay (PLA) (Sigma-Aldrich) fluorescence method was applied to detect in situ protein–protein interaction following the manufacturer's instructions. Third instar larval eye discs expressing proteins of interests were used in this assay. Rabbit anti-V5 (1:1,000, AB3792, Millipore) and mouse anti-GFP (1:500, 4C9, DSHB) antibodies were used to assess interactions between TER94 and Mu2. Rabbit anti-Myc (1:1,000, GTX109636, GeneTex) and mouse anti-GFP (1:500, 4C9, DSHB) antibodies were used to assess interactions between p53 and Mu2. Images were acquired with a Zeiss LSM 800 confocal microscope.

**EdU incorporation assay.** To determine whether cells re-entry cell cycle, Click-iT® EdU (Invitrogen) was used following the manufacturer's instructions. Larval or adult retinas were dissected in PBS and then incubated in S2 medium only (control) or S2 medium with 20 µM EdU for 1.5 h. After incubation, the tissues were fixed with 4% paraformaldehyde in PBS and permeabilize with 0.5% Triton® X-100. Mouse anti-Lamin (1:20, ADL67.10, DSHB) antibody was used to label the nucleus. Zeiss LSM-510 confocal microscope was used for collecting the images.

**Gamma irradiation.** Flies were irradiated with an open beam Cobalt-60 gamma source for the total dose of 1 Gy at the dose rate of 5.412 Gy/h. Control adult flies were mock irradiated.

**Quantification of a tandem GFP-mCherry-Atg8a reporter.** GFP-mCherry-Atg8a structures were analyzed using ImageJ as previously described[84]. Briefly, after feature extraction of puncta on the raw images, the GFP and mCherry intensity in each pixel across the center of the puncta was measured in arbitrary units (a.u.). The mean fluorophore intensity of the control group in an experimental set was used to define the gating thresholds of GFP and mCherry. Plot profile was used to visualize mCherry and GFP intensity from several Atg8a-positive structures to reveal dynamics of the reporter in the tested conditions; high intensity of both fluorophores indicates autophagosomes, high mCherry and low GFP indicates fusion with the lysosome as GFP is quenched in this organelle, and low mCherry reflects degradation within the lysosome. Quantification of percentage of puncta in each quadrant and the ratiometric analysis were derived from three independent experiments.

**Statistics and reproducibility.** GraphPad Prism 9 was used to analyze the statistical significance of all the data in this study. For the comparison of the relative nuclear cross-section size shown in Fig. 3b, the number of independent nuclei measured are 85 (LacZ), 68 (Derlin-1), and 46 (Derlin-1$^{\Delta SHP}$). Values represent mean ± SE. One-way ANOVA with Bonferroni's multiple comparison test compared to LacZ. ***$p$ < 1.0e−15 (Derlin-1); $p$ > 0.9999, ns (Derlin-1$^{\Delta SHP}$). For the comparison of nuclear TER94 intensity shown in Fig. 3b, the number of

independent nuclei measured are 40 for all tested genotypes. Values represent mean ± SE. One-way ANOVA with Bonferroni's multiple comparison test compared to LacZ. ***$p$ < 1.0e−15 (Derlin-1); $p$ = 0.3997, ns (Derlin-1$^{\Delta SHP}$). ***$p$ < 1.0e−15 by comparing Derlin-1 and Derlin-1$^{\Delta SHP}$. For the comparison of γH2AV intensity shown in Fig. 3e, the number of independent nuclei measured are 58 (0 day, LacZ), 159 (0 day, LacZ), 95 (2 day, LacZ), 162 (2day, TER94$^{K2A}$), 56 (5 day, LacZ), 88 (5 day, TER94$^{K2A}$), 62 (8 day, LacZ), and 73 (8 day, TER94$^{K2A}$). Values represent mean ± SE. One-way ANOVA with Bonferroni's multiple comparison test compared to LacZ at different time points. $p$ > 0.9999, ns (0 day); ***$p$ < 1.0e−15 (2day); ***$p$ < 1.0e−15 (5 day); ***$p$ < 1.0e−15 (8 day). For the comparison of γH2AV intensity shown in Fig. 3h, the number of independent nuclei measured are 246 for both L/D and D/D conditions. Values shown represent mean ± SE (Student's $t$ test, two-tailed, ***$p$ < 1.0e−15). For the comparison of the relative nuclear cross-section size shown in Fig. 3h, the number of independent nuclei measured are 329 for both L/D and D/D conditions. Values shown represent mean ± SE (Student's $t$ test, two-tailed, ***$p$ < 1.0e−15). For the comparison of γH2AV intensity shown in Fig. 3i, the number of independent nuclei measured are 320 for both Rh1 > TER94$^{K2A}$ + LacZ and Rh1 > TER94$^{K2A}$ + hSOD1. Values shown represent mean ± SE (Student's $t$ test, two-tailed, ***$p$ = 5.3e−10). For the comparison of the relative nuclear cross-section size shown in Fig. 3i, the number of independent nuclei measured are 236 for both Rh1 > TER94$^{K2A}$ + LacZ and Rh1 > TER94$^{K2A}$ + hSOD1. Values shown represent mean ± SE (Student's $t$ test, two-tailed, ***$p$ = 2.4e−4).

For RNAi screen shown in Fig. 4a, the number of independent nuclei measured are 211 (LacZ), 121 (Bon), 125 (Blm), 116 (NHEJ1), 127 (CG2990), 94 (XRCC4), 108 (CG32756), 123 (CG5825), 111 (CG5872), 104 (CG9272), 128 (dgrn), 108 (ERCC1), 93 (GEN), 116 (HDM), 114 (Ku80), 121 (Irbp), 80 (L3MBT), 175 (Mu2), 100 (mms4), 112 (Mei-9), 101 (Mre11), 93 (MBD-R2), 100 (Mei-41), 128 (mus81), 110 (Mus101), 192 (Mus205), 103 (Mus301), 84 (Mus304), 170 (Mus312), 113 (Mus308), 104 (CG9272), 99 (NBS), 92 (Ogg1), 115 (Okr), 117 (p53), 56 (Rad50), 119 (Rad51D), 118 (RPS3), 99 (Spn-A), 122 (Spn-B), 121 (Spn-D), 115 (Swm1), 115 (Slx1), 117 (Tos), 99 (TOP3), 102 (Thd1), 123 (Xpc), 122 (Tefu), 93 (XRCC2), and 108 (Lig4). For the comparison of the relative nuclear cross-section size shown in Fig. 4c, the number of independent nuclei measured are 126 (LacZ), 277 (eGFP-mu2), 152 (eGFP-mu2 + TER94$^{WT}$), and 261 (eGFP-mu2 + TER94 RNAi). Values represent mean ± SE. One-way ANOVA with Bonferroni's multiple comparison test compared to LacZ. ***$p$ < 1.0e−15 (eGFP-mu2); $p$ = 0.9685, ns (eGFP-mu2 + TER94$^{WT}$); ***$p$ < 1.0e−15 (eGFP-mu2 + TER94 RNAi). Comparing to eGFP-mu2, ***$p$ = 1.5e−5 (eGFP-mu2 + TER94$^{WT}$); ***$p$ < 1.0e−15 (eGFP-mu2 + TER94 RNAi). ***$p$ < 1.0e−15 by comparing eGFP-mu2 + TER94$^{WT}$ and eGFP-mu2 + TER94 RNAi. For the comparison of the nuclear eGFP intensity shown in Fig. 4d, the number of independent nuclei measured are 95 (LacZ), 267 (eGFP-mu2), 148 (eGFP-mu2 + TER94$^{WT}$), and 202 (eGFP-mu2 + TER94 RNAi). Values represent mean ± SE. One-way ANOVA with Bonferroni's multiple comparison test compared to LacZ. ***$p$ < 1.0e−15 (eGFP-mu2); ***$p$ = 2.9e−12 (eGFP-mu2 + TER94$^{WT}$); ***$p$ < 1.0e−15 (eGFP-mu2 + TER94 RNAi). Comparing to eGFP-mu2, ***$p$ = 3.2e−6 (eGFP-mu2 + TER94$^{WT}$); *$p$ = 0.0158 (eGFP-mu2 + TER94 RNAi). ***$p$ = 2.5e−12 by comparing eGFP-mu2 + TER94$^{WT}$ and eGFP-mu2 + TER94 RNAi. For the comparison of nuclear volume shown in Fig. 4e, the number of independent nuclei measured are 39 (LacZ), 74 (eGFP-mu2), 45 (eGFP-mu2 + TER94$^{WT}$), and 52 (eGFP-mu2 + TER94 RNAi). Values represent mean ± SE. One-way ANOVA with Bonferroni's multiple comparison test compared to LacZ. ***$p$ = 2.9e−7 (eGFP-mu2); $p$ = 0.1015, ns (eGFP-mu2 + TER94$^{WT}$); ***$p$ < 1.0e−15 (eGFP-mu2 + TER94 RNAi). Comparing to eGFP-mu2, *$p$ = 0.0115 (eGFP-mu2 + TER94$^{WT}$); ***$p$ = 0.0001 (eGFP-mu2 + TER94 RNAi). ***$p$ = 7.4e−10 by comparing eGFP-mu2 + TER94$^{WT}$ and eGFP-mu2 + TER94 RNAi. For the comparison of γH2AV intensity shown in Fig. 4g, the number of independent nuclei measured are 168 (LacZ), 134 (eGFP-mu2), 268 (eGFP-mu2 + TER94$^{WT}$), and 114 (eGFP-mu2 + TER94 RNAi). Values represent mean ± SE. One-way ANOVA with Bonferroni's multiple comparison test compared to LacZ. ***$p$ < 1.0e−15 (eGFP-mu2); ***$p$ = 2.5e−9 (eGFP-mu2 + TER94$^{WT}$); ***$p$ < 1.0e−15 (eGFP-mu2 + TER94 RNAi). Comparing to eGFP-mu2, ***$p$ < 1.0e−15 (eGFP-mu2 + TER94$^{WT}$); ***$p$ < 1.0e−15 (eGFP-mu2 + TER94 RNAi). ***$p$ < 1.0e−15 by comparing eGFP-mu2 + TER94$^{WT}$ and eGFP-mu2 + TER94 RNAi.

For the comparison of the relative nuclear cross-section size shown in Fig. 5a, the number of independent nuclei measured are 98 (LacZ), 80 (Mu2 + LacZ), and 110 (Mu2 + p53-RNAi). Values represent mean ± SE. One-way ANOVA with Bonferroni's multiple comparison test compared to LacZ. ***$p$ < 1.0e−15 (Mu2 + LacZ); ***$p$ = 7.4e−7 (Mu2 + p53-RNAi). ***$p$ < 1.0e−15 by comparing Mu2 + LacZ and Mu2 + p53-RNAi. In Fig. 5b, the number of independent nuclei measured are 219 (TER94$^{K2A}$ in $p53^{+/+}$ background) and 146 (TER94$^{K2A}$ in $p53^{-/-}$ background). Values shown represent mean ± SE (Student's $t$ test, two-tailed, ***$p$ < 1.0e−15). For the comparison of the relative nuclear cross-section size shown in Fig. 5c, the number of independent nuclei measured are 96 (eGFP-Mu2; $p53^{+/+}$), 98 (eGFP-Mu2; $p53^{-/-}$), 108 (eGFP-Mu2; $p53^{-/-}$ + 6XMyc-p53A), and 116 (eGFP-Mu2; $p53^{-/-}$ + 6XMyc-p53B). Values represent mean ± SE. One-way ANOVA with Bonferroni's multiple comparison test compared to eGFP-Mu2; $p53^{+/+}$. **$p$ = 0.0082 (eGFP-Mu2; $p53^{-/-}$); **$p$ = 0.0060 (eGFP-Mu2; $p53^{-/-}$ + 6XMyc-p53A); $p$ = 0.0838, ns (eGFP-Mu2; $p53^{-/-}$ + 6XMyc-p53B). Compared to eGFP-Mu2; $p53^{-/-}$, ***$p$ = 5.5e−10 (eGFP-Mu2; $p53^{-/-}$ + 6XMyc-p53A); $p$ >

0.9999, ns (eGFP-Mu2; $p53^{-/-}$ + 6XMyc-p53B). ***$p$ = 2.2e−8 by comparing eGFP-Mu2; $p53^{-/-}$ + 6XMyc-p53A and eGFP-Mu2; $p53^{-/-}$ + 6XMyc-p53B. For the comparison of nuclear volume shown in Fig. 5c, the number of independent nuclei measured are 37 (eGFP-Mu2; $p53^{+/+}$), 53 (eGFP-Mu2; $p53^{-/-}$), 37 (eGFP-Mu2; $p53^{-/-}$ + 6XMyc-p53A), and 46 (eGFP-Mu2; $p53^{-/-}$ + 6XMyc-p53B). Values represent mean ± SE. One-way ANOVA with Bonferroni's multiple comparison test compared to eGFP-Mu2; $p53^{+/+}$. $p$ = 0.4272, ns (eGFP-Mu2; $p53^{-/-}$); *$p$ = 0.0483 (eGFP-Mu2; $p53^{-/-}$ + 6XMyc-p53A); $p$ > 0.9999, ns (eGFP-Mu2; $p53^{-/-}$ + 6XMyc-p53B). Compared to eGFP-Mu2; $p53^{-/-}$, ***$p$ = 2.9e−5 (eGFP-Mu2; $p53^{-/-}$ + 6XMyc-p53A); $p$ > 0.9999, ns (eGFP-Mu2; $p53^{-/-}$ + 6XMyc-p53B). ***$p$ = 0.0004 by comparing eGFP-Mu2; $p53^{-/-}$ + 6XMyc-p53A and eGFP-Mu2; $p53^{-/-}$ + 6XMyc-p53B. In Fig. 5g, the number of independent nuclei measured are 77 (LacZ), 60 (6XMyc-p53A + LacZ), and 46 (6XMyc-p53A + TER94$^{K2A}$). Values represent mean ± SE. One-way ANOVA with Bonferroni's multiple comparison test compared to LacZ. $p$ = 0.9654, ns (6XMyc-p53A + LacZ); ***$p$ < 1.0e−15 (6XMyc-p53A + TER94$^{K2A}$). ***$p$ < 1.0e−15 by comparing 6XMyc-p53A + LacZ and 6XMyc-p53A + TER94$^{K2A}$.

For the comparison of GFP intensity of GFP-Atg8a shown in Fig. 6c, 5 eyes from each genotype were measured. Values represent mean ± SE. One-way ANOVA with Bonferroni's multiple comparison test compared to TER94$^{K2A}$ + GFP-Atg8a. **$p$ = 0.0012 (TER94$^{K2A}$ + GFP-Atg8a + p53-RNAi); **$p$ = 0.0010 (TER94$^{K2A}$ + GFP-Atg8a + Mu2-RNAi). For the comparison of GFP intensity of GFP-p62, 10 (TER94$^{K2A}$ + GFP-p62), 8 (TER94$^{K2A}$ + GFP-p62 + p53-RNAi), and 4 (TER94$^{K2A}$ + GFP-Atg8a + Mu2-RNAi) eyes were measured. Values represent mean ± SE. One-way ANOVA with Bonferroni's multiple comparison test compared to TER94$^{K2A}$ + GFP-p62. ***$p$ = 2.7e−9 (TER94$^{K2A}$ + GFP-p62 + p53-RNAi); ***$p$ = 1.6e−8 (TER94$^{K2A}$ + GFP-p62 + Mu2-RNAi). For the comparison of the tandem reporter shown in Fig. 6f, i, three independent experiments similar to the representative plots of Fig. 6e, h, respectively, were performed and values are presented in mean ± SE. In Fig. 6f, one-way ANOVA with Bonferroni's multiple comparison test compared to the first quadrant of DMSO (I): ***$p$ = 6.0e−9 (Rapamycin), **$p$ = 0.0013 (Chloroquine), and ***$p$ = 2.5e−6 (TER94$^{K2A}$); second quadrant of DMSO (II): ***$p$ = 2.1e−11 (Rapamycin), $p$ > 0.9999, ns (Chloroquine), and $p$ > 0.9999, ns (TER94$^{K2A}$); third quadrant of DMSO (III): $p$ > 0.9999, ns (Rapamycin), **$p$ = 0.0024 (Chloroquine), and ***$p$ = 1.4e−4 (TER94$^{K2A}$); fourth quadrant of DMSO (IV): $p$ > 0.9999, ns (Rapamycin), $p$ > 0.9999, ns (Chloroquine), and $p$ > 0.9999, ns (TER94$^{K2A}$). In Fig. 6i, Student's $t$ test, two-tailed compared to LacZ in each quadrant (Mu2-I, ***$p$ = 8.5e−4; Mu2-II, $p$ = 0.2199, ns; Mu2-III, **$p$ = 0.0014; Mu2-IV, $p$ = 0.1894, ns). For the comparison of GFP/mCherry ratio shown in Fig. 6g, j, the number of puncta measured are 41 (DMSO), 41 (Rapamycin), 47 (Chloroquine), and 44 (TER94$^{K2A}$). Values represent mean ± SE. One-way ANOVA with Bonferroni's multiple comparison test compared to DMSO. ***$p$ < 1.0e−15 (TER94$^{K2A}$); ***$p$ < 1.0e−15 (Chloroquine); ***$p$ < 1.0e−15 (Rapamycin). Comparing to TER94$^{K2A}$; $p$ > 0.9999, ns (Chloroquine); ***$p$ < 1.0e−15 (Rapamycin). ***$p$ < 1.0e−15 by comparing Chloroquine and Rapamycin. In Fig. 6j, 48 (LacZ) and 56 (Mu2) puncta were measured. Values represent mean ± SE (Student's $t$ test, two-tailed, ***$p$ = 2.2e−9). For the comparison of the relative nuclear cross-section size shown in Fig. 6k, the number of independent nuclei measured are 95 (LacZ), 58 (Mu2 + LacZ), 77 (Mu2 + 6XMyc-p53A), and 111 (Mu2 + 6XMyc-p53B). Values represent mean ± SE. One-way ANOVA with Bonferroni's multiple comparison test compared to LacZ. *$p$ = 0.0468 (Mu2 + LacZ); ***$p$ < 1.0e−15 (Mu2 + 6XMyc-p53A); *$p$ = 0.0110 (Mu2 + 6XMyc-p53B). Comparing to Mu2 + LacZ, ***$p$ = 3.5e−11 (Mu2 + 6XMyc-p53A); ***$p$ = 5.4e−7 (Mu2 + 6XMyc-p53B). Comparing to Mu2 + 6XMyc-p53A, ***$p$ < 1.0e−15 (Mu2 + 6XMyc-p53B).

**Reporting summary**. Further information on research design is available in the Nature Research Reporting Summary linked to this article.

## Data availability
All data generated or analyzed during this study are included in this published article (and its supplementary information files). The raw data underlying Figs. 1c, d, f, 2b, 3b, e, h–i, 4a, c–e, g–k, 5a–e, g, 6c, e–k and Supplementary Figs. 1b, 2b, d, 6b, c, 8b, d, 9b, 11 are available in the Source Data file provided with this paper. All unique materials generated are readily available from the authors. Source data are provided with this paper.

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

## Acknowledgements

We like to thank Ta-Fu Wang, Chien-Lun Huang, Jia-Chi Wang, and Liang-Yi Lin for assisting with the nucleus size quantification. We thank Drs. Darren Williams, Brian Calvi, Guang-Chao Chen, Bloomington Drosophila Stock Center, Kyoto Stock Center, Vienna *Drosophila* RNAi Center, and Fly Core Taiwan for generously providing fly strains. We thank the Image Core of the Brain Research Center, the Confocal Imaging Core, and the Isotope Division of the Nuclear Science and Technology Development Center at National Tsing Hua University for assistance with confocal microscopy and Cobalt-60 irradiation. We are grateful to Drs. Jui-Chou Hsu, Guang-Chao Chen, and Tim Tully for their helpful comments and discussion regarding this work. This work was supported by grants from the Ministry of Science and Technology of Taiwan (106-2811-B-007-008 and 109-2311-B-007-003), and the Higher Education Sprout Project, funded by the Ministry of Science and Technology and the Ministry of Education of Taiwan.

## Author contributions

Ya-Chu Chang and Tzu-Kang Sang conceived and designed the experiments. Ya-Chu Chang, Yu-Xiang Peng, Po-Hua Yu, Pei-Shin Liang, Ting-Yi Huang, Chao-Jie Shih, and Tzu-Kang Sang performed the experiments. Ya-Chu Chang, Yu-Xiang Peng, Po-Hua Yu, Pei-Shin Liang, Ting-Yi Huang, Chao-Jie Shih, Li-An Chu, and Tzu-Kang Sang analyzed the data. Ya-Chu Chang, Henry C Chang, and Tzu-Kang Sang wrote the paper.

## Competing interests

The authors declare no competing interests.
