## [Peer Review File · Nature Communications]

REVIEWER COMMENTS

Reviewer #1 (Remarks to the Author):

In their manuscript 'VCP functions to maintain nuclear size by managing DNA damage-induced MDC1-p53-autophagy axis' Ya-Chu Chang and co-authors describe their functional analysis of impaired VCP function linking aberrant nuclear morphology to emerging DNA damage and activation of the respective response/repair factors MDC1 and p53 in *Drosophila melanogaster* (*D. melanogaster*).

VCP is a central factor of the ubiquitin-proteasome system mediating selective degradation of ubiquitylated substrate proteins, which has been functionally linked to a variety of essential cellular processes involving chromatin-associated degradation pathways. Throughout the manuscript the authors use tissue specific expression of dominant negative alleles of Ter94 (the *D. melanogaster* ortholog of VCP) in the eye to show that impaired VCP function results in an age-dependent increase in nuclear size, correlating with emerging hallmarks of proteotoxicity and DNA damage. The correlation between Ter94 impairment and nuclear size is observed in other post mitotic tissues as well. Using a candidate approach to identify DNA repair factors responsible to mediate nuclear size increase, the authors identify Mu2 (the *D. melanogaster* ortholog of human MDC1), a well described initiating factor of the DNA double-strand break response. Interestingly, overexpression of Mu2 is sufficient to trigger nuclear size increase, while depletion of Mu2 suppresses nuclear morphology defects in the Ter94 mutant background. Accordingly, the authors propose that Mu2 is the crucial substrate of Ter94 in this context. This conclusion is supported by increased Mu2 protein levels upon Ter94 inhibition and colocalization in immune-stained tissue. The authors end their manuscript by presenting data implying that increased Mu2 levels upon Ter94 inhibition result in stabilized p53A isoform, which in turn results in perturbed autophagy.

The manuscript is written in a clear and conclusive way. The presented data is convincing and supports the conclusions drawn by the authors. With regards to the first part of the manuscript, the reviewer completely follows the authors description and conclusion that impaired Ter94 activity results in increased nuclear size, correlating with emergence of DNA damage and Mu2-dependent response. Identifying Mu2 as a substrate protein of Ter94 in DNA damage response is the strongest conclusion of this part of the manuscript. This conclusion is supported by a recent study published in *Genes and Development*, implying a role of RNF4 and VCP regulating MDC1 at the chromatin (Garvin et al., 2019, doi: 10.1101/gad.321125.118). With regards to the second part, linking p53A activity to perturbed autophagy remains largely hypothetical and requires further experimental validation. In particular, how autophagic degradation is functionally involved in the described process is not covered by the data. Based on the broad scientific interest regarding the regulation of DNA-damage response and the regulatory role of post-translational modifications, the manuscript can be of potential interest for publication in *Nature Communications*. However, I strongly recommend to address all critical points listed below.

Major points of concern

1) The authors' hypothesis that perturbed autophagy is causally connected to nuclear size increase needs experimental validation. First, the increase in Atg8 and p62 abundance/foci can be interpreted as a failure in autophagy execution, as the authors propose. However, these phenotypes could also display an increase in autophagy induction, possibly as a compensatory response to proteotoxicity upon Ter94 inhibition. In order to clarify in how far autophagy is affected, the authors need to validate their conclusion with an assay allowing analysis of autophagic flux, not steady state levels of autophagy markers. Second, it remains elusive in how far autophagy is involved in nuclear expansion, in the context of Ter94 impairment and in general. Does activation or inactivation of autophagy modulate nuclear size and other Ter94-linked phenotypes (e.g. DNA damage)? Furthermore, it remains undefined how the damage response is connected to autophagy regulation. Does Mu2 overexpression perturb autophagy, as it elicits nuclear expansion? Does overexpression of p53A suffice to perturb autophagy? In order to allow any conclusion concerning these points the authors

should further substantiate their hypotheses with experimental data.

2) While the reviewer appreciates the authors' conclusion that impaired Ter94 activity results in emergence of DNA-damage and elevated Mu2 levels/activity, it remains unclear where the double-strand breaks originate from in a post-mitotic tissue. In Figure 4a the authors show that depletion of DNA repair factors per se does not in any case result in increased nuclear size (at least in the Ter94 impaired background which is shown). This implies that the bare absence of effective DNA repair does not cause nuclear expansion. Presumably, a unique feature related to impaired Ter94 function links both aspects. Figure 4d shows that Ter94 overexpression suppresses the load of DNA damage caused by Mu2 overexpression, however, the Mu2 protein levels are not noticeably affected by Ter94 expression. Here, the authors provide correlation data of Mu2 signal intensity and nuclear size but no quantification of Mu2 is shown.

Additional points:

3) The authors use nuclear expansion as a readout in their analyses. It remains unclear however, what this increase in nuclear size indicates functionally. In this regard it would be informative to monitor nuclear size under stress conditions including DNA damage induction (activating Mu2/p53) but also proteotoxicity. Such analyses would facilitate a more conclusive view on nuclear expansion and what it indicates with regards to cellular physiology. Along the same line, additive to Ter94 inhibition, do cellular stresses (DNA damage, proteotoxicity) modulate the nuclear expansion phenotype?

4) The authors claim that nuclear expansion is an age-dependent phenomenon in the context of Ter94 inhibition. As they use Gal4-driven expression of dominant negative Ter94, I was wondering whether they can exclude an effect based on expression levels based on the Gal4 drivers. Are the Gal4- driver lines constantly expressed at identical level throughout the period as analysis? Furthermore, if the phenotype is age-dependent, it should be ruled out how far Mu2 overexpression/depletion affect tissue ageing in order to exclude secondary effects on phenotype manifestation due to modulation of the pace of tissue ageing.

5) Figure 1e: It does not become obvious were the homozygous cell clone is identified.

6) Figure 3: The figure should support that Ter94 function is crucial in the nucleus, however, neither in fluorescence images nor in the quantification, a nuclear enrichment upon Derlin-1 overexpression becomes obvious.

Reviewer #2 (Remarks to the Author):

Chang et al. report a new function for Drosophila p53, in TER94-dependent maintenance of nuclear size and functional autophagy. TER94 is an ATPase needed for subcellular segregation and degradation of ubiquitinated proteins. The authors show that expression of an inactive version of TER94 (K2A) results in increased levels of Ub-proteins and autophagy markers in post mitotic cells of the adult fly, with an accompanying increase in nuclear size. In a focused RNAi screen, depletion of a number of DDR proteins rescued the nuclear size defects. Overexpression of one of these, Mu2/MDC1, phenocopies the effect of TER94-K2A and further studies suggest the former is a substrate of TER94. The authors identify p53 as acting downstream of Mu2 to impair autophagy, to result in increased nuclear size. While the scope of the work is interesting and potentially suitable for Nature Communications, some of the conclusions are not supported by the data and there are possible alternate explanations for the findings. Specifically:

1. What is measured throughout the manuscript is nuclear cross-sectional area and not nuclear size. The authors should relate the first to the second, for example, by taking measurements in 3D of confocal images, which can be done in Image J. This needs not be done for all datasets but for representative nuclei of key datasets. This is because changes in nuclear shape without changes in volume, can produce increased cross-sectional area.

2. In general, data panels that show TER94 mutant clones (e.g. Fig. 1e) do not support the conclusions. These panels are supposed to show mutant clones with large nuclei. The arrow in 1e points to a single mutant nucleus that is GFP-and large. But there are lots of GFP- cells in the same field that have normal nuclei. This goes against the authors' conclusions. Further, such clonal data should be quantified as well (similar to Fig. 1c).

3. Fig. 3 on DNA damage is not convincing, for the following reasons.

3.1. Gamma-H2Av appears in both lacZ control and K2A panels in 3d. The signal may be specific in larval eye discs but does not appear to be in the adult eye. The correlation with nuclear size increase is also not convincing in the images shown and is only about ~50% over control in the graph. Related to 2, Fig. 3f shows a number of GFP- nuclei that are small AND display g-H2AV signal.

3.2. Fig.3g-PCNA signal is indeed higher in the experimental samples, but why is there signal outside the nucleus? PCNA should be nuclear. Is the antibody specific? Or are we seeing PCNA aggregates in the cytoplasm (which would be interesting)?

3.3. When there is more g-H2AV signal, the authors conclude that repair is impaired. But this result could also be because more damage is induced while repair is normal. Related, what is causing DNA damage in Rh1>TER94-K2A eyes? The cells are post-mitotic, so DNA replication cannot be the source.

4. The data with UAS-reporters for autophagy should be confirmed with antibodies. This is because when the authors see a difference with Mu2 or p53 RNAi co-expression, this could be a consequence of titrating limited GAL4 with 3 versus 2 UAS-transgenes. Antibodies to Atg8 and p62 have been published (PMID: 24667416). GFP reporters are used widely in the fly autophagy field, but these can also show differences from the endogenous protein (same paper).

5. Increased Atg8 and p62 after TER94-K2A expression is taken as defective autophagy (line 206). But these signals could result even if autophagy is active but the system is overwhelmed by the accumulation of Ub-proteins so that fusion to the lysosome becomes a bottleneck. Can the authors rule out this possibility?

6. Related to 5, the 'rescue of autophagy' by Mu2 or p53 RNAi (Fig. 6) could be because the load on autophagy has been reduced and thus the bottleneck is no longer an issue, rather than because DDR proteins play a regulatory role on autophagy as the model suggests. If so, reducing other (non Mu2) substrates of TER94 should 'rescue autophagy'. Has the authors tested this possibility?

7. Overall, the data could be explained by a simpler model in which TER94-K2A results in accumulation of its substrate Mu2. Mu2 stabilizes downstream DDR proteins, overwhelming the system. In this explanation, there is no need to invoke DNA damage or defective repair (the data for which is not convincing anyways). Can the authors rule out this possibility?

Reviewer #3 (Remarks to the Author):

This manuscript shows an effect of the transitional endoplasmic reticulum ATPase VCP (TER94 in *Drosophila*) on nuclear size. Expression of dominant negative TER94 caused increases in nuclear size in photoreceptor cells. These changes correlated with levels of higher levels of ubiquitinated nuclear proteins. TER94 apparently functions in the nucleus and in the cytoplasm and, although not discussed, TER94 may be a suppressor of ubiquitin-linked proteolysis. Loss of TER94 function causes the accumulation of MDC1 which is used for checkpoint mediated cell cycle arrest in response to DNA damage. MDC1-overexpression also increased nuclear size. TER94 decreases levels of MDC1 which is a stabilizer of p53 and this is proposed as a mechanism upstream of nuclear size increases.

The phenotype that is discovered is an increase in nuclear size and the mechanism for this is explored. Other interactions and effects are shown in this manuscript but the headline, the title, and the conclusions are about nuclear size. Nuclear size is tightly controlled and the mechanisms for this control have received much interest. They have been studied in many species including *Drosophila* since this is a fundamental problem in biology. While this manuscript focuses on nuclear size, it does not show much interest in the biology of nuclear size regulation. Previously published work and ideas about nuclear size control are not discussed and there is no effort to relate the experiments and results presented here to that previous work or to previously discovered mechanisms. Among these missing ideas and mechanisms are that nuclear size is regulated by DNA content, by nuclear/cytoplasmic ratios, by nucleocytoplasmic transport, by transcription levels, and/or by the LINC complex that mechanically connects the nucleus to the cytoskeleton.

This would be more appropriate for a general audience interested in nuclear architecture if it showed a better awareness of the big picture, a consideration of previous results on nuclear size, and experiments informed by that previous work. The experiments move down a hierarchical path of factors that lead to an isoform of p53 upstream of nuclear size increase. The proposed mechanism up through p53 is interesting, plausible, and supported by the experiments. It is interesting that p53 affects nuclear size but the mechanism by which this might happen is not elucidated. The final linkage is not made. This is important if the focus of the paper is nuclear size.

Some of the early text concerning apoptosis versus autophagy was puzzling. It was not clear why apoptosis was assumed to cause nuclear enlargement rather than nuclear fragmentation. Statements about autophagy recur throughout the text. These allusions to autophagy, as distinct from the ubiquitin proteasome system, were unclear. One would think that autophagy refers to the degradation of cytoplasmic structures and cells. Perhaps, there is background not presented and documenting a role for autophagy in this *Drosophila* model? There is a section that discusses "autophagic flux" which I would have thought referred to the degradation of material in autophagosomes. That is, however, not directly addressed. Only some factors that could be involved in such a process, like Atg8 and p62, are addressed. Is there some confirming experiment that could fill this gap?

Overall, these are interesting experiments and results that have some gaps as a story about nuclear size.

Specific Comments:

1. For readers (like me) who do not work on compound eyes, the authors might want to define "rhabdomere" and explain why their nuclear areas were normalized to rhabdomeral size in some experiments rather than something like cytoplasmic area or cell area. The cartoon of Figure 1A does not help much in clarifying this. Explaining the basic system would be useful for general readers. Interest in nuclear size regulation goes beyond the fly world.

2. Judging from Figure 1, expression of dominant negative TER94 caused nuclear shape changes. Is this figure representative? This result seems different than supplemental Figure 1. If the image is representative, why was this not pursued? The introduction begins by talking about changes in nuclear size and shape in aging and disease but shape is not really presented. If measured shape does not change while size does, that is a significant finding and could be documented.

3. The authors should discuss how much expression and activity their hypomorphic mutant TER94K15502 does have.

4. The very first paragraph discusses aging. Are there increases in nuclear size in aging? In what systems? If there is some point they want to make about aging they should discuss this.

5. What is the IMBPF mentioned? Is this "inclusion body myopathy with early-onset Paget disease

and frontotemporal dementia"? Are there implications in these results for frontotemporal dementia or Paget disease?

6. Line 134, anti-lamin antibodies label the "nuclear lamina" not the "nuclear envelope". We expect that these are close together most of the time, but it is an important distinction.

7. Are enlarged nuclei a "defect" as in Line 220? Are there consequences on nuclear function?

8. Line 106. What is "the known TER94 role" in this regard. A little background with references would have helped.

9. Figure 3a and 3b present the effect of Derlin-1 expression on TER94 subcellular distribution and nuclear size. Each condition has one measurement of a line through two nuclei, "randomly selected". There should be a better way of quantifying this in $n =$ more than 2 nuclei.

10. In Figure 4a, the range of nuclear sizes is surprisingly large.

11. Figure 5 panels e and f are mislabeled.

We thank the reviewers for the constructive comments and suggestions. To address them, we have performed additional experiments and included some of the results in the revision. We believe the revised manuscript has improved significantly and hope the reviewers will agree.

Please find our point-by-point response to the Reviewers below:

REVIEWER COMMENTS

Reviewer #1 (Remarks to the Author):

*In their manuscript 'VCP functions to maintain nuclear size by managing DNA damage-induced MDC1-p53-autophagy axis' Ya-Chu Chang and co-authors describe their functional analysis of impaired VCP function linking aberrant nuclear morphology to emerging DNA damage and activation of the respective response/repair factors MDC1 and p53 in *Drosophila melanogaster* (*D. melanogaster*).*

*VCP is a central factor of the ubiquitin-proteasome system mediating selective degradation of ubiquitylated substrate proteins, which has been functionally linked to a variety of essential cellular processes involving chromatin-associated degradation pathways. Throughout the manuscript the authors use tissue specific expression of dominant negative alleles of Ter94 (the *D. melanogaster* ortholog of VCP) in the eye to show that impaired VCP function results in an age-dependent increase in nuclear size, correlating with emerging hallmarks of proteotoxicity and DNA damage. The correlation between Ter94 impairment and nuclear size is observed in other post mitotic tissues as well. Using a candidate approach to identify DNA repair factors responsible to mediate nuclear size increase, the authors identify Mu2 (the *D. melanogaster* ortholog of human MDC1), a well described initiating factor of the DNA double-strand break response. Interestingly, overexpression of Mu2 is sufficient to trigger nuclear size increase, while depletion of Mu2 suppresses nuclear morphology defects in the Ter94 mutant background. Accordingly, the authors propose that Mu2 is the crucial substrate of Ter94 in this context. This conclusion is supported by increased Mu2 protein levels upon Ter94 inhibition and colocalization in immune-stained tissue. The authors end their manuscript by presenting data implying that increased Mu2 levels upon Ter94 inhibition result in stabilized p53A isoform, which in turn results in perturbed autophagy.*

*The manuscript is written in a clear and conclusive way. The presented data is convincing and supports the conclusions drawn by the authors. With regards to the first part of the manuscript, the reviewer completely follows the authors description and conclusion that impaired Ter94 activity results in increased nuclear size, correlating with emergence of DNA damage and Mu2-dependent response. Identifying Mu2 as a substrate protein of Ter94 in DNA damage response is the strongest conclusion of this part of the manuscript. This conclusion is supported by a recent study published in *Genes and Development*, implying a role of RNF4 and VCP regulating MDC1 at the chromatin (Garvin et al., 2019, doi: 10.1101/gad.321125.118). With regards to the second part, linking p53A activity to perturbed autophagy remains largely hypothetical and requires further experimental validation. In particular, how autophagic degradation is functionally involved in the*

described process is not covered by the data. Based on the broad scientific interest regarding the regulation of DNA-damage response and the regulatory role of post-translational modifications, the manuscript can be of potential interest for publication in *Nature Communications*. However, I strongly recommend to address all critical points listed below.

Major points of concern

1) *The authors' hypothesis that perturbed autophagy is causally connected to nuclear size increase needs experimental validation. First, the increase in Atg8 and p62 abundance/foci can be interpreted as a failure in autophagy execution, as the authors propose. However, these phenotypes could also display an increase in autophagy induction, possibly as a compensatory response to proteotoxicity upon Ter94 inhibition. In order to clarify in how far autophagy is affected, the authors need to validate their conclusion with an assay allowing analysis of autophagic flux, not steady state levels of autophagy markers. Second, it remains elusive in how far autophagy is involved in nuclear expansion, in the context of Ter94 impairment and in general. Does activation or inactivation of autophagy modulate nuclear size and other Ter94-linked phenotypes (e.g. DNA damage)? Furthermore, it remains undefined how the damage response is connected to autophagy regulation. Does Mu2 overexpression perturb autophagy, as it elicits nuclear expansion? Does overexpression of p53A suffice to perturb autophagy? In order to allow any conclusion concerning these points the authors should further substantiate their hypotheses with experimental data.*

Regarding the first point, we agree with the reviewers that this is a critical experiment and have included the tandem GFP-mCherry-Atg8a reporter to measure the autophagy flux. This ratiometric analysis takes advantage of the fact that the GFP fluorescence in this dual color chimera is quenched by the acidic environment during autophagosome-lysosome fusion. Based on this principle, this tandem reporter can reveal the states of autophagosomes at the autophagosome initiation stage (yellow puncta; I), the lysosomal fusion stage (red puncta; GFP quenched; II), and the lysosomal degradation stage (small red puncta; fusion degraded; III). This tandem reporter, like GFP-Atg8, showed diffused nuclear signal in *Rh1>TER94^{K2A}*. We thus focused on puncta of certain sizes across the samples. We showed that, compared to *Rh1>LacZ*, a significant proportion of the puncta in *Rh1>TER94^{K2A}* appeared to be at Stage I, suggesting that TER94 dysfunction arrests the progression of autophagy. A similar defect was observed in Mu2 overexpressing retinas. These results are shown in the revised Fig. 6.

Regarding the second point, while we do not have functional data demonstrating perturbation of autophagy could directly modulate the nuclear size, we have data showing manipulations of TER94, Mu2, and p53A affect both nuclear size and autophagy. These correlations suggest these two processes might be linked. Mu2 overexpression increases ref(2)p accumulation (Fig. 6k in this revision) and elevates the percentage of Stage I autophagosomes in tandem reporter analysis (Figure 6h-j), suggesting that it affects autophagy, as well as nuclear expansion. We have included a new data showing overexpression of p53A alone can perturb nuclear size. p53A overexpression alone can induce nuclear expansion, when p53 is accumulated to a sufficient level. Although *Rh1-GAL4* is active in all of the outer R cells, *Rh1>p53A* only enlarges a small number of nuclei because p53A level is low in most of the cells

(Fig. 5g). As p53A is labile, its capability to enlarge nuclei is greatly enhanced by the stabilization by TER94 dysfunction or Mu2 accumulation (Fig. 5c, g). Importantly, we found that p53A overexpression does not increase γ H2AV levels (Supplementary Fig. 11 of this revision), indicating that DNA damages and nuclear size increase can be uncoupled. We propose that in the presence of unrepaired DNA damages, accumulated Mu2 stabilizes p53A, which then causes nuclear expansion.

2) While the reviewer appreciates the authors' conclusion that impaired Ter94 activity results in emergence of DNA-damage and elevated Mu2 levels/activity, it remains unclear where the double-strand breaks originate from in a post-mitotic tissue. In Figure 4a the authors show that depletion of DNA repair factors per se does not in any case result in increased nuclear size (at least in the Ter94 impaired background which is shown). This implies that the bare absence of effective DNA repair does not cause nuclear expansion. Presumably, a unique feature related to impaired Ter94 function links both aspects. Figure 4d shows that Ter94 overexpression suppresses the load of DNA damage caused by Mu2 overexpression, however, the Mu2 protein levels are not noticeably affected by Ter94 expression. Here, the authors provide correlation data of Mu2 signal intensity and nuclear size but no quantification of Mu2 is shown.

We have approached this issue from two perspectives. First, we have performed a new experiment aimed to ensure that γ H2AV detects DSB with high fidelity. We show that wild type R cells exhibit a transient γ H2AV increase in response to a short exposure to γ -radiation. While we cannot formally exclude the possibility that γ H2AV reacts with other epitopes, the kinetics of this response, along with the elevated γ H2AV signals in hydrogen peroxide-treated eye discs (Fig. 3c), suggests that γ H2AV is highly specific for DSB. In addition, the transience of response demonstrates that cells have the capability to quickly repair induced DNA lesions, in contrast to TER94^{K2A}-expressing R cells. This result is shown in Supplementary Fig. 8 of revision.

In addition, we have added two new experiments monitoring γ H2AV levels in *Rh1>TER94^{K2A}* flies with reduced exposures to environmental mutagens. First, we reared *Rh1>TER94^{K2A}* flies in the dark, which should reduce DNA damages caused by radiation and UV light. Compared to *Rh1>TER94^{K2A}* grown in the regular 12h light/dark cycle, those raised in constant dark showed suppressed γ H2AV accumulation and reduced nuclear expansion. We also expressed human superoxide dismutase in *Rh1>TER94^{K2A}*, and found the co-expression of this reactive oxygen radical scavenger reduced γ H2AV levels. These results suggest that increased γ H2AV levels in *Rh1>TER94^{K2A}* represent accumulation of unrepaired DNA damages from environmental genotoxic agents. These results are shown in the revised Fig. 3.

Moreover, using p53A overexpression, we now have a better understanding of the relationship between DNA damages and nuclear size enlargement. The phenotype of p53A overexpression alone shows that the stabilization of p53A is a key event for nuclear enlargement, and DNA damages and nuclear enlargement can be uncoupled. Thus, the significance of DNA damages appears to induce Mu2 accumulation, which stabilizes p53A, thereby enhancing the ability of p53A to enlarge nuclei.

The quantification of eGFP-Mu2 levels in TER94 modulated conditions for Fig. 4b is now shown in panel 4d. A quantification (see PbP Figure 1 below) for an independent dataset presented in the original 4d (now in Fig. 4f) was performed. In both cases, Mu2 signal is reduced by TER94 overexpression and enhanced by TER94 RNAi.

PbP Figure 1. TER94 negatively regulate the level of Mu2. Quantification of the eGFP intensity from *Rh1>LacZ*, *Rh1>eGFP-Mu2*, *Rh1>eGFP-Mu2, TER94^{WT}*, or *Rh1>eGFP-Mu2, TER94-RNAi*. 115-167 nuclei from ≥ 3 independent eyes are measured in each group. Values shown represent mean \pm SE. ** $p < 0.01$, *** $p < 0.001$ (one-way ANOVA with Bonferroni's multiple comparison test).

Additional points:

3) *The authors use nuclear expansion as a readout in their analyses. It remains unclear however, what this increase in nuclear size indicates functionally. In this regard it would be informative to monitor nuclear size under stress conditions including DNA damage induction (activating Mu2/p53) but also proteotoxicity. Such analyses would facilitate a more conclusive view on nuclear expansion and what it indicates with regards to cellular physiology. Along the same line, additive to Ter94 inhibition, do cellular stresses (DNA damage, proteotoxicity) modulate the nuclear expansion phenotype?*

As TER94^{K2A} expression causes accumulation of ubiquitinated protein in the nuclei, it is possible that this accumulation affects the nuclear/cytoplasmic transport. As for the rationale of nuclear enlargement in *Rh1>TER94^{K2A}*, it is possible that as post-mitotic cells accumulate unrepaired DNA damages, the nuclear size expand to slow down other cellular processes (by reducing their effective concentrations), thereby allowing DDR machinery more opportunities to repair the DNA lesions. Alternatively, chromatins surrounding DSBs in multiple systems have been shown to exhibit increased mobility, possibly allowing the recruitment of DNA repair proteins. An increase in nuclear size should provide damaged chromatins more space to move. It is worth pointing out that DNA damages have not been identified as a contributing factor for nuclear size control, and this work could provide a framework to address this important question. While we are also interested in understand the physiological rationale of nuclear expansion in response to DNA damages, we did not include this in the text, as we feel this issue is beyond the scope of this paper.

4) *The authors claim that nuclear expansion is an age-dependent phenomenon in the context of Ter94 inhibition. As they use Gal4-driven expression of dominant negative Ter94, I was wondering whether they can exclude an effect based on expression levels based on the Gal4 drivers. Are the Gal4- driver lines constantly expressed at identical level throughout the period as analysis? Furthermore, if the phenotype is age-dependent, it should be ruled out how far Mu2 overexpression/depletion affect tissue ageing in order to exclude secondary effects on phenotype manifestation due to modulation of the pace of tissue ageing.*

We have independently verified the *Rh1>TER94^{K2A}* phenotypes (e.g. enlarged nuclei, increased FK2 staining, accumulation of γ H2AV) with clones of a TER94 loss-of-function mutation. We have also performed luciferase activity to track Rh1 activity over time. Rh1 expression was lower in 1-day-old extract (~40% of the stabilized level), but stabilized as the flies aged (see PbP Figure 2 below). We are confident that the age-dependent nuclear size increase is not a result of *Rh1-GAL4* being more active in older flies.

Manipulations of three genes (ie TER94^{K2A}, Mu2 overexpression, and p53A overexpression) all cause progressive nuclear enlargement, and it seems unlikely to us that all three achieve this by affecting tissue aging. Furthermore, we have obtained a new data that p53A overexpression can bypass the requirement of increased DNA damages to enlarge nuclei. This observation, in combination with the result that Mu2 stabilizes p53A, suggests the age-dependence of the nuclear enlargement in TER94^{K2A} reflects the extent of accumulated DNA damages. In the absence of TER94 function (or the presence of Mu2 overexpression), unrepaired DNA damages accumulate with age, which increases the level of stabilized p53A.

PbP Figure 2. The expression kinetics of Rh1-GAL4 driver. Quantification of the illumination of firefly luciferase driven by *Rh1-GAL4* at indicated ages. N=3.

5) *Figure 1e: It does not become obvious were the homozygous cell clone is identified.*

We have repeated the experiment and replaced the old image with a new one, showing one homozygous TER94^{K15502} R cell, labeled by the absence of GFP signal in rhabdomere (Fig. 1f). To illustrate the cell positions, we marked the boundaries of outer R cells with dash lines to illustrate the clone position better.

6) *Figure 3: The figure should support that Ter94 function is crucial in the nucleus, however, neither in fluorescence images nor in the quantification, a nuclear enrichment upon Derlin-1 overexpression becomes obvious.*

We respectfully disagree on this point. We believe that our image data adequately showed overexpressing Derlin-1 depleted endogenous TER94 from the nuclei. In addition, overexpression of a Derlin-1 mutant and the rescue with TER94^{WT} support this conclusion. Nevertheless, to bolster this point, we have added panel in the revised Fig. 3b, showing a quantitative comparison of nuclear size and the relative subcellular distribution of TER94.

Reviewer #2 (Remarks to the Author):

Chang et al. report a new function for Drosophila p53, in TER94-dependent maintenance of nuclear size and functional autophagy. TER94 is an ATPase needed for subcellular segregation and degradation of ubiquitinated proteins. The authors show that expression of an inactive version of TER94 (K2A) results in increased levels of Ub-proteins and autophagy markers in post mitotic cells of the adult fly, with an accompanying increase in nuclear size. In a focused RNAi screen, depletion of a number of DDR proteins rescued the nuclear size defects. Overexpression of one of these, Mu2/MDC1, phenocopies the effect of TER94-K2A and further studies suggest the former is a substrate of TER94. The authors identify p53 as acting downstream of Mu2 to impair autophagy, to result in increased nuclear size. While the scope of the work is interesting and potentially suitable for Nature Communications, some of the conclusions are not supported by the data and there are possible alternate explanations for the findings. Specifically:

1. What is measured throughout the manuscript is nuclear cross-sectional area and not nuclear size. The authors should relate the first to the second, for example, by taking measurements in 3D of confocal images, which can be done in Image J. This needs not be done for all datasets but for representative nuclei of key datasets. This is because changes in nuclear shape without changes in volume, can produce increased cross-sectional area.

We thank the reviewer's suggestion. In this revision, we have repeated the key datasets, including TER94^{K2A}, Mu2, p53 isoforms, and the corresponded control, and acquired the Z stacks of anti-Lamin sections to reconstitute the 3D volume using Avizo (Thermo Fisher Scientific). These results, presented in Fig. 1d, 4b and 5c, are in agreement with the measurement of cross-section of R cell nuclei. We have also re-labeled the axis to "Relative nuclear cross-section size" to make this distinction.

2. In general, data panels that show TER94 mutant clones (e.g. Fig. 1e) do not support the conclusions. These panels are supposed to show mutant clones with large nuclei. The arrow in 1e points to a single mutant nucleus that is GFP-and large. But there are lots of GFP- cells in the same field that have normal nuclei. This goes against the authors' conclusions. Further, such clonal data should be quantified as well (similar to Fig. 1c).

We have replaced the mosaic image (Fig. 1f) with a new one, showing one homozygous TER94^{K15502} R cell, labeled by the absence of GFP in the rhabdomere. To illustrate the position of these cells, we marked the boundaries of outer R cells with dash lines. In addition, we have included a quantitative comparison of nuclear sizes between normal and homozygous TER94^{K15502} mutant cells (Fig. 1f).

3. Fig. 3 on DNA damage is not convincing, for the following reasons.
3.1. Gamma-H2Av appears in both lacZ control and K2A panels in 3d. The signal may be specific in larval eye discs but does not appear to be in the adult eye. The correlation with nuclear size increase is also not convincing in the images shown and is only about ~50% over control in the graph. Related to 2, Fig. 3f shows a number of GFP- nuclei that are small AND display g-H2AV signal.

As for Fig. 3d, we commonly observed some γ H2AV background in our experiments. To demonstrate that γ H2AV is specific in detecting DNA lesions in the adult retina, we have included a new Supplementary Fig. 8, showing γ H2AV level transiently increases in wild type adult retina exposed to radiation.

We have also used PCNA to independently demonstrate the increased DNA lesions in $Rh1>TER94^{K2A}$.

We have replaced the mosaic image (Fig. 3f) with a new one, showing one homozygous $TER94^{K15502}$ R cell, labeled by the absence of GFP in the rhabdomere. The boundaries of outer R cells were marked with dash lines. Consistent with the observation that expressing dominant-negative $TER94$ ($Rh1>TER94^{K2A}$) shows increased γ H2AV, this single cell clone presents more robust signal compared to the neighboring cells.

3.2. Fig.3g-PCNA signal is indeed higher in the experimental samples, but why is there signal outside the nucleus? PCNA should be nuclear. Is the antibody specific? Or are we seeing PCNA aggregates in the cytoplasm (which would be interesting)?

We also notice PCNA signal in the cytoplasm. As the control and the newly emerged $TER94^{K2A}$ adult of the same experiment that were processed in parallel did not show evident PCNA staining, it is unlikely that the signals were non-specific. While cytoplasmic PCNA aggregates have been reported in different cell types and may have ramification in influencing apoptosis (<https://doi.org/10.1084/jem.20092241>; <https://doi.org/10.1371/journal.pone.0117546>), we did not include this discussion in the text, as we feel that this is beyond the scope of this paper, as we used PCNA mainly as an independent DDR marker to verify the increase in γ H2AV.

3.3. When there is more γ -H2AV signal, the authors conclude that repair is impaired. But this result could also be because more damage is induced while repair is normal. Related, what is causing DNA damage in $Rh1>TER94-K2A$ eyes? The cells are post-mitotic, so DNA replication cannot be the source.

To test whether this increased γ H2AV signal could be caused by more induced DNA damages alone, we monitored γ H2AV levels in adult retina exposed to radiation. Unlike $Rh1>TER94^{K2A}$, the radiation-induced γ H2AV increase was transient, indicating that wild type cells are capable of quickly repair DNA lesions. In contrast, $Rh1>TER94^{K2A}$ exposed to radiation showed sustained increase (Supplementary Fig. 8). This observation suggests the sustained increase in γ H2AV level is caused by a disruption in DNA repair.

4. The data with UAS-reporters for autophagy should be confirmed with antibodies. This is because when the authors see a difference with Mu2 or p53 RNAi co-expression, this could be a consequence of titrating limited GAL4 with 3 versus 2 UAS-transgenes. Antibodies to Atg8 and p62 have been published (PMID: 24667416). GFP reporters are used widely in the fly autophagy field, but these can also show differences from the endogenous protein (same paper).

We agree and have performed experiment with immunostaining for endogenous ref(2)p (the fly homolog of p62) and Atg8. The results confirmed that the knockdown

of either p53 or Mu2 suppresses TER94^{K2A}-induced accumulation of autophagic markers (Supplementary Fig. 12 in this revision).

5. Increased Atg8 and p62 after TER94-K2A expression is taken as defective autophagy (line 206). But these signals could result even if autophagy is active but the system is overwhelmed by the accumulation of Ub-proteins so that fusion to the lysosome becomes a bottleneck. Can the authors rule out this possibility?

This is a tricky question: how to distinguish between autophagy defective at a specific step versus a bottleneck in a functional but overwhelmed autophagy? The best argument we have that the nuclear enlargement is not caused by a saturation of the system with ubiquitinated proteins is Mu2 overexpression, which while capable of inducing enlarged nuclei, does not increase FK2 staining (Supplementary Fig. 10 in this revision). This new result suggests it is not the accumulation of ubiquitinated proteins per se that causes nuclear enlargement. Rather, it is the accumulation of a specific protein (ie Mu2) that results in nuclear enlargement. It is likely that the ubiquitinated proteins observed in TER94^{K2A} include Mu2, and we include another new data showing co-expression of Mu2 in Rh1>TER94^{K2A} significantly increases FK2 signals (Supplementary Fig. 10 in this revision). Over-expressed Mu2 in Rh1>Mu2, while not being ubiquitinated, mimics the nuclear accumulation of Mu2 proteins in TER94^{K2A}.

6. Related to 5, the 'rescue of autophagy' by Mu2 or p53 RNAi (Fig. 6) could be because the load on autophagy has been reduced and thus the bottleneck is no longer an issue, rather than because DDR proteins play a regulatory role on autophagy as the model suggests. If so, reducing other (non Mu2) substrates of TER94 should 'rescue autophagy'. Has the authors tested this possibility?

As this question is similar to 5, please see our response to the previous question.

7. Overall, the data could be explained by a simpler model in which TER94-K2A results in accumulation of its substrate Mu2. Mu2 stabilizes downstream DDR proteins, overwhelming the system. In this explanation, there is no need to invoke DNA damage or defective repair (the data for which is not convincing anyways). Can the authors rule out this possibility?

We agree with the reviewer, and we believe that we have direct evidence for this scenario with two new data. In this revision, we show that Mu2 overexpression can uncouple increased FK2 from enlarged nuclei (Supplementary Fig. 10 in this revision). Similarly, p53A overexpression can uncouple DNA damages from enlarged nuclei (Fig. 5g and Supplementary Fig. 11 in this revision). These observations are more consistent with TER94, Mu2, and p53A acting in a sequential manner than each causing a stress (eg accumulation of excessive DNA damages or ubiquitinated proteins) to overwhelm the system. In our model, DNA damages per se do not increase nuclear size, but DNA damages have an important function of stabilizing p53A, which affects nuclear size. Direct p53A overexpression mimics the stabilization of p53A, thus bypassing the need of accumulation of DNA damages.

Reviewer #3 (Remarks to the Author):

This manuscript shows an effect of the transitional endoplasmic reticulum ATPase VCP (TER94 in Drosophila) on nuclear size. Expression of dominant negative TER94 caused increases in nuclear size in photoreceptor cells. These changes correlated with levels of higher levels of ubiquitinated nuclear proteins. TER94 apparently functions in the nucleus and in the cytoplasm and, although not discussed, TER94 may be a suppressor of ubiquitin-linked proteolysis. Loss of TER94 function causes the accumulation of MDC1 which is used for checkpoint mediated cell cycle arrest in response to DNA damage. MDC1-overexpression also increased nuclear size. TER94 decreases levels of MDC1 which is a stabilizer of p53 and this is proposed as a mechanism upstream of nuclear size increases.

The phenotype that is discovered is an increase in nuclear size and the mechanism for this is explored. Other interactions and effects are shown in this manuscript but the headline, the title, and the conclusions are about nuclear size. Nuclear size is tightly controlled and the mechanisms for this control have received much interest. They have been studied in many species including Drosophila since this is a fundamental problem in biology. While this manuscript focuses on nuclear size, it does not show much interest in the biology of nuclear size regulation. Previously published work and ideas about nuclear size control are not discussed and there is no effort to relate the experiments and results presented here to that previous work or to previously discovered mechanisms. Among these missing ideas and mechanisms are that nuclear size is regulated by DNA content, by nuclear/cytoplasmic ratios, by nucleocytoplasmic transport, by transcription levels, and/or by the LINC complex that mechanically connects the nucleus to the cytoskeleton.

We thank the Reviewer for the comment and apologize for the oversight. In the revised manuscript, we have modified the text significantly in Introduction and Discussion to include more background on nuclear size control. Specifically, we have included analysis of N/C ratio control from other systems. The text (on page 3) now reads: It has long been appreciated that cells keep a relatively constant karyoplasmic ratio, i.e., the proportion of nuclear volume and cytoplasmic volume, implying maintaining such a physiological consistency has important roles. Indeed, aberrant nuclear size has served as a pathological hallmark in several types of malignant tumors. However, it is unclear whether the change in nuclear size has a direct role in causing diseases or it represents a byproduct of another cellular defect that causes diseases. To address this, it is critical to understand the regulatory processes, as well as the cellular stresses, that impact nuclear size.

Studies from multiple systems have revealed that nuclear size is proportionally scaled to the cytoplasmic volume. This reliance on cytoplasmic volume suggests that nucleus can sense determinants for nuclear size control reside outside the nucleus. The import of nuclear lamina proteins through the nuclear pore complexes (NPCs) has been implicated in the nucleus growth during interphase. The availability of ER membrane, the source for reassembled nuclear envelope at the end of cell division, also constrains the nucleus size and shape after mitosis. Recent genetic screens have further implicated nucleocytoplasmic transport, RNA processing, and nucleocytoskeletal interactions as key mechanisms for controlling nucleocytoplasmic ratio. In addition, in the Discussion (page 18), we have added a paragraph discussing a possible link between our findings and LINC complexes. LINC complexes have been implicated in increasing the mobility of chromatin bearing DSB. Our results provide

additional evidence supporting the connection between DNA damages and nuclear size control. The text now reads: LINC complexes, implicated in N/C control, comprise inner and outer nuclear membrane proteins and function to bridge the nuclear lamina and the cytoskeleton. In yeast and mammalian cells, chromatin associated with DSBs exhibited increased mobility and this phenomenon requires LINC complexes, raising a potential link between chromatin repair to nuclear architecture. Our analysis suggests p53 may participate in this or a similar process, possibly functioning as a sensor of unrepaired DNA damages for machinery that regulates the nuclear size. Indeed, 53BP1, a p53 binding protein, decorates the damaged foci, although whether p53 has a direct role in this process remains to be shown.

Moreover, we have included a discussion of the FK2 signal increase in TER94 deficient cells. *Rh1>TER94^{K2A}* shows accumulation of ubiquitinated proteins in nuclei. Similar defect has also been observed in cells treated with drugs that block nucleocytoplasmic transport, resulting in nuclear size increase. The observation of accumulation of nuclear proteins in TER94^{K2A} suggests this imbalance could be a signal for cellular stress. In support of this, we further suggest that the identities of these accumulated ubiquitinated proteins in TER94^{K2A} appear to be specific to those functioning in DDR.

This would be more appropriate for a general audience interested in nuclear architecture if it showed a better awareness of the big picture, a consideration of previous results on nuclear size, and experiments informed by that previous work. The experiments move down a hierarchical path of factors that lead to an isoform of p53 upstream of nuclear size increase. The proposed mechanism up through p53 is interesting, plausible, and supported by the experiments. It is interesting that p53 affects nuclear size but the mechanism by which this might happen is not elucidated. The final linkage is not made. This is important if the focus of the paper is nuclear size.

We have included a new data showing overexpression of p53A alone is sufficient to increase nuclear size (Fig. 5g in this revision). This phenotype has a low penetrance because p53A is unstable without the stabilization by Mu2. In the presence of Mu2 accumulation, the ability of p53A to expand nuclear size is strongly enhanced (Fig. 5c, e in this revision). Hence, our work now paints a clear picture of how TER94 dysfunction leads to nuclear expansion. In the absence of nuclear TER94 function, Mu2 is not cleared, resulting in accumulation of unrepaired DNA damages and stabilization of p53A.

Based on a previous report of p53A's impact on autophagy (ref. 55 in the text), we speculated that stabilization of p53A might disrupt autophagy, which could affect the removal of damaged nuclear components.

Some of the early text concerning apoptosis versus autophagy was puzzling. It was not clear why apoptosis was assumed to cause nuclear enlargement rather than nuclear fragmentation. Statements about autophagy recur throughout the text. These allusions to autophagy, as distinct from the ubiquitin proteasome system, were unclear. One would think that autophagy refers to the degradation of cytoplasmic structures and cells. Perhaps, there is background not presented and documenting a

role for autophagy in this Drosophila model? There is a section that discusses “autophagic flux” which I would have thought referred to the degradation of material in autophagosomes. That is, however, not directly addressed. Only some factors that could be involved in such a process, like Atg8 and p62, are addressed. Is there some confirming experiment that could fill this gap?

We agree with the reviewer that nuclear fragmentation, not nuclear enlargement, is associated with apoptosis. Given the known TER94 role in apoptosis, we wanted to formally exclude apoptosis as a cause for nuclear enlargement.

While eukaryotic cells harness the ubiquitin proteasome system (UPS) and autophagy to maintain homeostasis, the general view is that UPS processes short-lived proteins, whereas autophagy removes long-lived proteins and protein complexes, or damaged organelles. Both UPS and autophagy require ubiquitin-conjugation of their substrates (FK2 staining); however, the nuclear size increase occurs way ahead of UPS disruption in *Rh1>TER94^{K2A}, UAS-CD3δ-YFP* (CD3δ-YFP is a reporter for ER-associated degradation as well as an UPS substrate) (Supplementary Fig. 4 in this revision). In addition, TER94^{K2A} and Mu2 overexpression lead to LC3/Atg8a and p62 accumulation, in addition to nuclear expansion. We thus focus on autophagy.

As for the autophagy’s role in processing nuclear components, a previous study in mammalian cells has shown that DNA, Histone H1, Lamin A/B, LINC component Emerin, and γH2AX are associated the LC3-marked autophagosomes (<https://doi.org/10.4161/auto.8901>), suggesting autophagy is required for nucleus homeostasis. Studies from budding yeast revealed the so-called nucleophagy (or piecemeal micronucleophagy), which can pinch off a portion of nuclear envelope to remove nuclear components ([doi: 10.1091/mbc.E02-08-0483](https://doi.org/10.1091/mbc.E02-08-0483); <https://doi.org/10.1038/nature14506>). Furthermore, cellular stresses inside the nucleus, such as DNA damage, can induce autophagy, and the process requires DNA checkpoint kinases like ATM and ATR, and its downstream effector kinase Chk2 (<https://doi.org/10.1073/pnas.1614364114>). As genotoxic induced Chk2 activation directly phosphorylates p53 to interfere Mdm2 binding and thereby stabilize p53 ([doi: 10.1126/science.287.5459.1824](https://doi.org/10.1126/science.287.5459.1824)), one could imagine that p53’s role in both autophagy inhibition and DNA damage response may be the link.

To confirm the change in autophagy flux is associated with TER94 dysfunction, we have included new experiments using a dual color Atg8a reporter (GFP-mCherry-Atg8a) to monitor the dynamics of autophagic process (Fig. 6d-j in this revision). The ratiometric analysis takes advantage of the fact that the GFP fluorescence in this dual color chimera is quenched by the acidic environment during autophagosome-lysosome fusion. Based on this principle, this tandem reporter can reveal the states of autophagosomes at the autophagosome initiation stage (yellow puncta; I), the lysosomal fusion stage (red puncta; GFP quenched; II), and the lysosomal degradation stage (small red puncta; fusion degraded; III). This tandem reporter, like GFP-Atg8, showed diffused nuclear signal in *Rh1>TER94^{K2A}*. We thus focused on puncta of certain sizes across the samples. We showed that, compared to *Rh1>LacZ*, a significant proportion of the puncta in *Rh1>TER94^{K2A}* appeared to be at Stage I, suggesting that TER94 dysfunction arrests the progression of autophagy. A

similar defect was observed in Mu2 overexpressing retinas. These results adhere the observed endogenous (antibody staining) and exogenous (GFP-Atg8a) markers.

Overall, these are interesting experiments and results that have some gaps as a story about nuclear size.

Specific Comments:

1. For readers (like me) who do not work on compound eyes, the authors might want to define “rhabdomere” and explain why their nuclear areas were normalized to rhabdomeral size in some experiments rather than something like cytoplasmic area or cell area. The cartoon of Figure 1A does not help much in clarifying this. Explaining the basic system would be useful for general readers. Interest in nuclear size regulation goes beyond the fly world.

We have included a sentence on page 6 to briefly describe the physiological function of rhabdomeres. It reads, “We measured the R cell nuclear sizes by quantifying antibody-labeled nuclear areas in confocal sections and normalizing them to the cross-section areas of rhabdomeres, cylindrical photo-sensory organelles present in all R cells, from the same cells.” As rhabdomeres extend throughout the depth of the retina, they are a convenient reference for comparing organelle size changes in fly photoreceptors.

2. Judging from Figure 1, expression of dominant negative TER94 caused nuclear shape changes. Is this figure representative? This result seems different than supplemental Figure 1. If the image is representative, why was this not pursued? The introduction begins by talking about changes in nuclear size and shape in aging and disease but shape is not really presented. If measured shape does not change while size does, that is a significant finding and could be documented.

As far as we can tell, expression of TER94^{K2A} affects nuclear size, but has no effect on the shape. In this revision, we have shown 3D rendering and measured the sphericity of TER94^{K2A}-expressing nuclei (Fig. 1d in this revision).

3. The authors should discuss how much expression and activity their hypomorphic mutant TER94K15502 does have.

The TER94^{K15502} is a lethal allele, containing a P-element transposon inserted in the TER94 5' untranslated region. Consistent with its fundamental functions, clones of TER94^{K15502} homozygous cells were end-divided and rare, which precludes us from quantifying the protein level in homozygous mutant cells carrying this allele. A Western below (PbP Figure 3) shows TER94^{K15502} heterozygotes (lane 4) express ~40% of TER94 proteins compared to wild type (lane 1). We chose this allele because excision of the P-element fully rescued the lethality, indicating that no other lethal mutation is on this TER94^{K15502}-containing chromosome (León and McKearin, <https://doi.org/10.1091/mbc.10.11.3825>).

PbP Figure 3. The expression level of *TER94*^{K15502} mutant. Western blot is probed with a customized anti-TER94 antibody to detect TER94 in lysates from *GMR-GAL4* alone (-), *GMR-GAL4>TER94^{WT}* (*TER94^{WT}*), *GMR-GAL4>TER94-RNAi* (*TER94-RNAi*), or *TER94^{K15502}* heterozygotes (*TER94^{-/+}*). The β -Tubulin blot serves as a loading control.

4. The very first paragraph discusses aging. Are there increases in nuclear size in aging? In what systems? If there is some point they want to make about aging they should discuss this.

In 1992, Pienta *et al.* measured the nuclear area of human skin fibroblasts from donors ranging from 1.2 to 96-year-old and found that the nuclei increase in size with age ([https://doi.org/10.1016/0047-6374\(92\)90140-9](https://doi.org/10.1016/0047-6374(92)90140-9)). The connection between nuclear shape and aging is stronger in literatures, and TER94 dysfunction appears to affect nuclear size, not shape. Thus we have removed references to aging throughout the text.

5. What is the IBMPFD mentioned? Is this “inclusion body myopathy with early-onset Paget disease and frontotemporal dementia”? Are there implications in these results for frontotemporal dementia or Paget disease?

Yes, IBMPFD is the dementia. The implication is that expression of the diseased allele does not cause nuclear enlargement, implying this phenotype does not contribute to the pathology. We have added a sentence on this on page 7.

6. Line 134, anti-lamin antibodies label the “nuclear lamina” not the “nuclear envelope”. We expect that these are close together most of the time, but it is an important distinction.

We thank the reviewer for pointing this out, and have replaced it with “lamina” (on page 6) in the revised manuscript.

7. Are enlarged nuclei a “defect” as in Line 220? Are there consequences on nuclear function?

We have changed the wording, and it now reads, “This correlation, along with the fact that the appearance of FK2 staining precedes nuclear enlargement, suggests the failure in removing ubiquitinated nuclear proteins ...”.

Regarding the impact of FK2 signal increase on other nuclear functions besides nuclear enlargement, we didn't look. As the cell size, the nuclear size, and the transcription levels appear to be correlated ([doi: 10.1083/jcb.128.4.467](https://doi.org/10.1083/jcb.128.4.467)), it is likely that transcription would be globally affected by TER94^{K2A} expression. The physiological rationale and impact of enlarged nuclei are interesting and important topics, and should be investigated. In this manuscript, we focused on elucidating the mechanistic link between DNA damages and nuclear size increase.

8. Line 106. What is “the known TER94 role” in this regard. A little background with references would have helped.

VCP/TER94 functions to extract ubiquitinated proteins from organelles and protein complexes for proteasome-mediated degradation, re-folding, or liberation. What is less clear is TER94 function in the nucleus. We have modified the sentence stating “Consistent with the many known VCP nuclear substrates, ubiquitinated proteins accumulate inside the nucleus of TER94 dysfunction cells” and citing Heidelberger et al.'s report on proteomic profiling of VCP substrates to make the sentence clear. We thank the reviewer's suggestion.

9. Figure 3a and 3b present the effect of Derlin-1 expression on TER94 subcellular distribution and nuclear size. Each condition has one measurement of a line through two nuclei, “randomly selected”. There should be a better way of quantifying this in n= more than 2 nuclei.

In the revision, we have added multiple measurements of TER94 subcellular localization, and the corresponded quantification and statistics are now included in the revised Fig. 3.

10. In Figure 4a, the range of nuclear sizes is surprisingly large.

The RNAi screen was conducted using lines from various sources. As these lines were generated with flies of different genetic backgrounds, it was expected to see a larger variation.

11. Figure 5 panels e and f are mislabeled.

We thank Reviewer for pointing this out, and have fixed this error.

REVIEWERS' COMMENTS

Reviewer #2 (Remarks to the Author):

The revised manuscript by Chang et al address most but not all my concerns as explained below.

1. Changes are now documented for nuclear volume and not just cross-sectional area, removing this concern.

2. New images in Fig. 1f are still confusing. The nucleus indicated with the yellow arrowhead is indeed large but the nucleus just below it is GFP- and about 4-fold smaller. Yet in the accompanying graph, the range of data points within the GFP- mutant group is only about 2-fold. How can that be? Aren't the data points for the image shown included in the graph? Is it because for each nucleus, only the largest cross-sectional area from different confocal slices is counted? This is not apparent from the Methods.

3. New data removed my concerns about gamma-H2Av staining.

4. New data removed my concerns about reporters versus staining for endogenous protein.

5 and 6. New data with the tandem reporter to monitor where in autophagy pathway the mutants/depletions blocks helps to address my concerns about whether the problem is in autophagy induction or resolution by fusion with the lysosome.

7. The authors seem to agree with my alternate interpretation of the data, yet they are still including DNA damage as the cause for nuclear enlargement in the title, the model figure, and the abstract. Their own data show that DNA damage is neither necessary (p53 or Mu2 overexpression) nor sufficient (environmental insults are found as the cause of DNA damage, yet nuclear size is normal by definition) for nuclear size control. Fig. 3h and I attempt to address the role of DNA damage experimentally but these are problematic. Culturing flies in the dark reduces environmental DNA damage as seen by gamma-H2Av stain in h and this is found to also reduce nuclear size, which fits with the author's model that DNA damage is the root cause of nuclear enlargement when TER94 is compromised. But the cells being studied are photoreceptors and they will not be stimulated in the dark. Changes in nuclear size may have more to do with whether they are being stimulated by light than DNA damage. Can this be ruled out? SOD1 expression data in Fig. 3I are not helpful because although the difference may be significant, the magnitude of change in nuclear size is too small to convince me that DNA damage is the cause of nuclear enlargement in this context.

Reviewer #3 (Remarks to the Author):

The authors have made significant changes in response to the first review, changes that clarify and strengthen the submission. In particular, previously described mechanisms contributing to nuclear size regulation are better considered in the text in relation to the results. The correlations between autophagy and nuclear size are not reduced to specific mechanisms and consequences on nuclear functions are not fully explored but there is enough here, there is a significant and clear story to tell.

REVIEWERS' COMMENTS

Reviewer #2 (Remarks to the Author):

The revised manuscript by Chang et al address most but not all my concerns as explained below.

1. Changes are now documented for nuclear volume and not just cross-sectional area, removing this concern.

2. New images in Fig. 1f are still confusing. The nucleus indicated with the yellow arrowhead is indeed large but the nucleus just below it is GFP⁻ and about 4-fold smaller. Yet in the accompanying graph, the range of data points within the GFP⁻ mutant group is only about 2-fold. How can that be? Aren't the data points for the image shown included in the graph? Is it because for each nucleus, only the largest cross-sectional area from different confocal slices is counted? This is not apparent from the Methods.

In the mosaic cluster shown in Figure 1f, only the cell indicated with the yellow arrowhead (a R5) is GFP⁻ (hence, TER94⁻). The other outer photoreceptors (R1, 2, 3, 4, and 6) are GFP⁺. In the revised Figure, we have labeled their associated rhabdomeres with "+" and "-" to make the genotypes more obvious.

3. New data removed my concerns about gamma-H2Av staining.

4. New data removed my concerns about reporters versus staining for endogenous protein.

5 and 6. New data with the tandem reporter to monitor where in autophagy pathway the mutants/depletions blocks helps to address my concerns about whether the problem is in autophagy induction or resolution by fusion with the lysosome.

7. The authors seem to agree with my alternate interpretation of the data, yet they are still including DNA damage as the cause for nuclear enlargement in **the title, the model figure, and the abstract**. Their own data show that DNA damage is neither necessary (p53 or Mu2 overexpression) nor sufficient (environmental insults are found as the cause of DNA damage, yet nuclear size is normal by definition) for nuclear size control.

We agree that DNA damage per se is not the cause of aberrant nuclear enlargement. In this revision, we have changed the word "induced" in the Title to "associated". The inclusion of the "DNA damage" phrase is to indicate that VCP regulates the nuclear size by influencing MDC1 and p53 levels in the context of DNA damage repair pathway.

We have changed the word "and" to "or" in the Abstract to indicate the fact that while loss of VCP function leads to MDC1 accumulation, it could be the responses associated with DNA damages that enlarge nuclei.

As for the model figure, we have removed the sentence about ATM phosphorylating H2AV in the Figure Legends, and have simplified the model panel in Figure 6 to emphasize more on the accumulation of DDR proteins and less on DNA damages. With these modifications, we believe the model figure accurately represents the sentence on page 19 (where the model figure is referenced), "VCP dysfunction causes aberrant accumulation of MDC1 on the DNA damage foci".

Fig. 3h and I attempt to address the role of DNA damage experimentally but these are problematic. Culturing flies in the dark reduces environmental DNA damage as seen by gamma-H2Av stain in h and this is found to also reduce nuclear size, which fits with the author's model that DNA damage is the root cause of nuclear enlargement when TER94 is compromised. But the cells being studied are photoreceptors and they will not be stimulated in the dark. Changes in nuclear size may have more to do with whether they are being stimulated by light than DNA damage. Can this be ruled out?

In principle, this can be addressed with a measurement of the nuclear size of *Rh1>TER94^{K2A}* flies reared in light/dark cycles in a *norpA* background. Removal of the *norpA* gene, a phospholipase C acting downstream of rhodopsin, disrupts phototransduction, and we can use this property to ask if the nuclear size difference in light/dark cycles depends on fly vision. However, obtaining *norpA* alleles and performing the measurements will take significantly longer than two weeks. Furthermore, as Covid-19 cases have recently spiked in Taiwan, the prospect of conducting this experiment in a timely manner remains uncertain.

The purpose of this experiment is to demonstrate the source for the DNA damages, indicated by γ H2AV staining, in *Rh1>TER94^{K2A}* flies. We have used two independent methods to minimize DNA damages, and tested their effect on nuclear size increase. While the effect of SOD1 overexpression, as pointed out by the Reviewer, is modest (see below), the results from these two approaches are consistent. That is, both methods decreased γ H2AV staining and suppressed *Rh1>TER94^{K2A}*-induced nuclear size increase. Thus, while we cannot experimentally rule out photoreceptor stimulation having a role, the SOD1 overexpression result argues for the importance of exposure to mutagens.

SOD1 expression data in Fig. 3I are not helpful because although the difference may be significant, the magnitude of change in nuclear size is too small to convince me that DNA damage is the cause of nuclear enlargement in this context.

We agree that the effect of SOD1 overexpression on *Rh1>TER94^{K2A}*-induced nuclear size increase is modest. However, the effect of SOD1 overexpression on γ H2AV staining in *Rh1>TER94^{K2A}* is also weak. Thus, the impact of SOD1 overexpression on these two *Rh1>TER94^{K2A}* phenotypes seems consistent. As

for the last point, we agree with the alternative interpretation suggested by the Reviewer, and have provided evidence with Mu2 and p53A overexpression that DNA damage itself is not the cause for nuclear enlargement.

Reviewer #3 (Remarks to the Author):

The authors have made significant changes in response to the first review, changes that clarify and strengthen the submission. In particular, previously described mechanisms contributing to nuclear size regulation are better considered in the text in relation to the results. The correlations between autophagy and nuclear size are not reduced to specific mechanisms and consequences on nuclear functions are not fully explored but there is enough here, there is a significant and clear story to tell.

We thank Reviewer for the suggestions, which help us improve the manuscript.